

# 1 Compositional balance should be considered in the mapping of soil
# 2 particle-size fractions using hybrid interpolators

Mo Zhang[1,2], Wenjiao Shi[1,3]
[1]Key Laboratory of Land Surface Pattern and Simulation, Institute of Geographic Sciences and Natural Resources Research,
Chinese Academy of Sciences, Beijing 100101, China
[2]School of Earth Sciences and Resources, China University of Geosciences, Beijing 100083, China
[3]College of Resources and Environment, University of Chinese Academy of Sciences, Beijing 100049, China
*Correspondence to:* Wenjiao Shi (shiwj@lreis.ac.cn), Institute of Geographic Sciences and Natural Resources Research,
Chinese Academy of Sciences. 11A, Datun Road, Chaoyang District, Beijing 100101, China.
**Abstract**. Digital soil mapping of soil particle-size fractions (PSFs) using log-ratio methods is a widely used technique. As a
hybrid interpolator, regression kriging (RK) is an alternative way to improve prediction accuracy. However, there is still a lack
of comparisons and recommendations when RK is applied for compositional data, and it is not known if the performance based
on different balances of isometric log-ratio (ILR) transformation is robust. Here, we compared the generalized linear model
(GLM), random forest (RF), and their hybrid patterns (RK) using different transformed data based on three ILR balances, with
29 environmental covariables (ECs) for the prediction of soil PSFs in the upper reaches of the Heihe River Basin, China. The
results showed that RF performed best, with more accurate predictions, but GLM produced a more unbiased prediction. For
the hybrid interpolators, RK was recommended because it widened the data ranges of the prediction values, and modified the
bias and accuracy for most models, especially for RF. Moreover, prediction maps generated from RK revealed more details of
the soil sampling points. For three ILR balances, different data distributions were produced. Using the most abundant
component of the compositional data as the first component of the permutations was not considered the right choice because
it produced the worst performance. Compared to the relative abundance of components, we recommend that the focus should
be on data distribution. This study provides a reference for the mapping of soil PSFs combined with transformed data at the
regional scale.
**1 Introduction**
Recently, spatial interpolation of soil particle-size fractions (PSFs) has become a focus of soil science researchers. More
accurately predicted soil PSFs could contribute to a better understanding of hydrological, physical, and environmental
processes (Delbari et al., 2011; Ließ et al., 2012; McBratney et al., 2002).
The characteristics of compositional data makes soil PSFs more impressive than other soil properties. Soil PSFs are usually
expressed as three components of discrete data – sand, silt, and clay, and carry only relevant percentage information. Soil
texture is classified as soil PSFs, which can be demonstrated on a ternary diagram (so-called soil texture triangle). The closure
system formed in this triangle is not Euclidean space, but is rather Aitchison space (i.e., the simplex) (Aitchison, 1986). Due





to "spurious correlations" (Pawlowsky-Glahn, 1984), traditional statistical methods based on the Euclidean geometry may
generate mistakes when dealing directly with soil PSF data (Filzmoser et al., 2009). The requirement for constant sum,
nonnegative, unbiased prediction is the key to spatial interpolation (Walvoort and de Gruijter, 2001). Data transformation is
crucial for compositional data from the simplex to the real space. Log ratio transformations play a significant role in
compositional data analysis, including the additive log-ratio (ALR), centered log-ratio (CLR) (Aitchison, 1986), and isometric
log-ratio (ILR) (Egozcue et al., 2003).

Although these three log-ratio methods have been widely applied to transform soil PSF data, different study area scales and

model selection should be considered when modeling. For local scale study areas, geostatistical models, i.e., ordinary kriging
(OK) and compositional kriging, combined with log-ratio transformed data, are sufficient to map spatial patterns, as shown in
our previous study (Wang and Shi, 2017). As another perspective, functional compositions combined with the kriging method
can also be applied to produce soil particle size curves (PSCs) (Menafoglio et al., 2014), providing an abundance of information.
This involves the use of complete and continuous information rather than discrete information, and soil PSFs can be extracted
from the predicted soil PSCs (Menafoglio et al., 2016a). Log-ratio transformations can also be combined with functional-
compositional data for the stochastic simulation of PSCs (Menafoglio et al., 2016b, Talska et al., 2018). For middle scale study
areas, outliers may lead to the overestimation of the variogram, resulting in prediction errors (Lark, 2000). Therefore, the
spatial interpolation should take robust variogram estimators into account to improve model performance (Lark, 2003). A
previous study proved that applying robust variogram estimators in log-ratio co-kriging significantly improved mapping
performance (Wang and Shi, 2018). For large scale study areas, geostatistical models are limited by the number of soil samples
and increased spatial variability. An increasing number of studies have concentrated on mapping soil PSFs using different
machine-learning models combined with ancillary data (i.e., environmental covariables, ECs) on a broad basin scale (Zhang
et al., 2020), national scale (Akpa et al., 2014), and even global scale (Hengl et al., 2017) using log-ratio transformed data.

Among these EC-combined models, linear, machine-learning, geostatistical models, and high accuracy surface modeling

(Yue et al., 2020) have been commonly used in middle or large-scale studies. Linear models, for example, the generalized
linear model (GLM) and multiple linear regression (MLR) have been used in soil PSF predictions with their flexibility and
interpretability (Lane, 2002; Buchanan et al., 2012). Many machine-learning models have been applied for the soil PSF
interpolation and soil texture classification. For example, tree learners, such as the random forest (RF), have been shown to be
advantageous due to their ability to handle overfitting and generate more realistic maps (Zhang et al., 2020). Furthermore,
regression kriging (RK), which has been proved to be a powerful and widely accepted method of soil mapping, can not only
combine ECs through its regression function, but it also improves model accuracy as a hybrid interpolator for some soil
properties, such as topsoil thickness and pH (Hengl et al., 2004; Keskin and Grunwald, 2018). However, the scope of the
comparison needs to be expanded to further explore the accuracy and predict compositional data using linear models, machine-
learning models, and other models combining RK (hybrid patterns).

In log-ratio methods, the ILR method performs better than ALR and CLR in both theory and in practice (Filzmoser and

Hron, 2009; Wang and Shi, 2018; Zhang et al., 2020). The ILR method eliminates model collinearity and preserves



advantageous properties such as isometry, scale invariance, and sub-compositional coherence, through its use of orthonormal
coordinate systems (i.e., balances) using a sequential binary partition (SBP) (Egozcue and Pawlowsky-Glahn, 2005). These
choices are not unique, multiple sets of ILR transformed data can be generated by permutations of components (different SBPs)
in the compositional data. The choice of an SBP can be based on prior expert knowledge, using a compositional biplot (Lloyd
et al., 2012) or variograms and cross-variograms (Molayemat et al., 2018). It has been proven in statistical science that different
results are obtained using different choices of ILR balances, and the option of a specific SBP for compositions is crucial for
the intended interpretation of coordinates (Fiserova and Hron, 2011). However, most soil science researchers have ignored this
point. Martins et al. (2016) reported that clay has been used as the denominator in the ALR method because it is typically the
most abundant component of compositions. Few studies have compared the different SBP options from the perspective of
accurate assessments and analyzed whether these differences are due to the general characteristics of specific data sets or log-
ratio transformations.
Therefore, based on our previous work, the objectives of this study were to: (i) compare the spatial prediction accuracy of
soil PSFs using a GLM and RF combined with ECs and ILR transformed data; (ii) determine whether hybrid interpolators
(GLMRK and RFRK) can improve the interpolation performance; and (iii) explore the distributions of different transformed
data and the variation law of precision based on different choices of SBP.
**2 Methods and materials**
**2.1 Study area**
The study area was the upper reaches of the Heihe River Basin (HRB), which is the source of the Heihe River and the central
area of runoff generation in the HRB. The elevation in this area ranges from 1640 to 5573 m (Fig. 1), and the climate is damp
and cold, being dominated by the Qilian Mountains. The mean annual rainfall in the study area is 350 mm, and the mean annual
temperature is lower than 4°C. Meadow and steppe are the dominant vegetation types. Grassland is the primary land-use type.
The main soil classes are frigid calcic soil in the southwest of the study area, with cold desert soil dominating the southeast,
while Castanozems and Sierozems are distributed in the north of the study area.

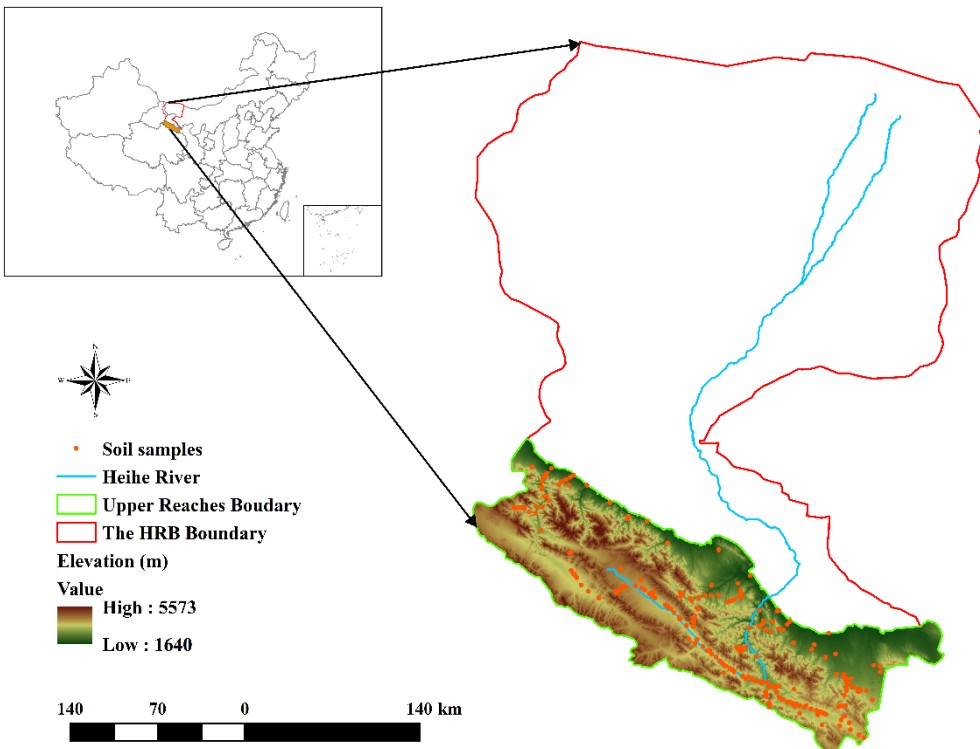


**Figure. 1.** The location, elevation, and soil samples on the upper reaches of the Heihe River Basin.


**2.2 Data collection and analysis**


**2.2.1 Soil PSF data**


A total of 262 soil samples were collected in the upper reaches of the HRB based on a purposive sampling strategy and were
used to characterize the spatial variability of soil PSFs at the regional scale (Fig. 1). The variability of soil formation factors,
such as elevation, soil type, vegetation class, and geomorphology of the upper reaches of the HRB was considered in soil
sample collection. The average of three mixed topsoil samples (approximate depth of 0–20 cm) was obtained to reduce the
noise of soil sample parameters, and a parallel sample was also measured. Subsequently, about 30 g of each soil sample was
air-dried, and chemical and physical analyses were conducted in the laboratory. Soil PSF information was obtained for the soil
samples using a Malvern Panalytical Mastersizer 2000, with less than 3% average measurement error.

**2.2.2 The selection of ECs**

There were 29 ECs considered in our study, including both continuous and categorical variables (Table S1.1). They followed
the principles of the SCORPAN model (McBratney et al., 2003). The continuous variables included the morphometry and
hydrologic characteristics of topographic properties, climatic and vegetative indices, and soil physical and chemical properties
(Yi et al., 2015; Song et al., 2016; Yang et al., 2016). The categorical variables included geomorphology, land use types, and


vegetation classes, which were transformed into raster with 1000 m resolution. Due to the intricate patterns of topography in
the upper reaches of the HRB, the variable of topographic properties dominated the ECs. The System for Automated
Geoscientific Analyses geographic information system (SAGA GIS, Conrad et al., 2015) was applied for a terrain analysis to
derive topographic variables using the 30 m resolution digital elevation model (DEM, http://www.gscloud.cn). A collinearity
test removed the redundant variables, and the topographic properties were then resampled to 1000 m. More details of the ECs
are provided in the Data Availability section.
**2.3 ILR transformation and SBP**
An orthonormal basis of ILR was chosen to isometrically project the compositions from $S^D$ (the simplex for the Aitchison
geometry) to $R^{D-1}$ (real space for the Euclidean geometry). The choice of a specific orthonormal basis for use on $S^D$ can be
explained by the SBP for the groups of compositions (Egozcue and Pawlowsky-Glahn, 2005). The choice of the construction
of coordinates (i.e., balances) between groups of compositions was calculated as follows:
$$z_k = \sqrt{\frac{r_k s_k}{r_k + s_k}} ln(\frac{(x_{i_1} x_{i_2} \dots x_{i_{r_k}})^{1/r_k}}{(x_{j_1} x_{j_2} \dots x_{j_{s_k}})^{1/s_k}}), \; k = 1, \dots, D - 1,$$ (1)
where $z_k$ refers to the balance between two groups; $i_1, i_2, \dots, i_{r_k}$ is the $r_k$ part of one group; and $j_1, j_2, \dots, j_{r_k}$ is the $s_k$
part of the other group. Therefore, in a stepwise manner, the balances contain all the relevant information of the compositions
in two groups. This can also be explained in a tabular form. For soil PSF data (D = 3), all three choices of the balance of SBPs
are shown in Table 1. The first component of the ILR contained all the information on soil PSFs, and the main difference in
the choice of balances for soil PSFs was the order of the three parts, i.e., the first order of the soil PSF component was used as
the numerator of the first ILR equation. In our study, three SBP balances, SBP1, SBP2, and SBP3, were transformed from the
original soil PSF data, and the orders of soil PSF data were $(sand, silt, clay)$, $(silt, clay, sand)$, and $(clay, sand, silt)$,
respectively. The transformation equations for the ILR can be derived from Eq. (1), and were defined as Eqs. (2) and (3). The
inverse equations for ILR were defined as Eqs. (4), (5), (6). The ILR transformation and its inverse were conducted using the
R package "compositions" (K. Gerald van den Boogaart and Raimon Tolosana, 2014).
$\mathbf{z} = (z_1, \dots z_{D-1}) = ILR(\mathbf{x})$, and for $i = 1, \dots, D - 1$ and component $x_i$, (2)
$$z_i = \sqrt{\frac{D-i}{D-i+1}} ln \frac{x_i}{\sqrt[D-i]{\Pi_{j=i+1}^D x_j}}.$$ (3)
$$Y(x_j) = \sum_{j=1}^D \frac{ILR(x_j)}{\sqrt{j \times (j+1)}} - \sqrt{\frac{j-1}{j}} \times ILR(x_j),$$ (4)
$ILR(x_0) = ILR(x_D) = 0,$ (5)
$$\overline{ILR}(x_j) = \frac{exp(Y(x_j))}{\sum_{j=1}^D exp(Y(x_j))}.$$ (6)
**Table 1** All choices of SBPs for soil PSF data (D = 3), the orders of soil PSFs data are $(sand, silt, clay)$, $(silt, clay, sand)$
and $(clay, sand, silt)$ for SBP1, SBP2 and SBP3.



| Groups | Step | Sand | Silt | Clay | r | s | Balance |
|--------|------|------|------|------|---|---|---------|
| SBP1 | 1 | + | - | - | 1 | 2 | Step1: $z_1 = \sqrt{\frac{2}{3}} ln \frac{sand}{\sqrt{silt \times clay}}$ |
|      | 2 | 0 | + | - | 1 | 1 | Step2: $z_2 = \sqrt{\frac{1}{2}} ln \frac{silt}{clay}$ |
| SBP2 | 1 | - | + | - | 1 | 2 | Step1: $z_1 = \sqrt{\frac{2}{3}} ln \frac{silt}{\sqrt{clay \times sand}}$ |
|      | 2 | - | 0 | + | 1 | 1 | Step2: $z_2 = \sqrt{\frac{1}{2}} ln \frac{clay}{sand}$ |
| SBP3 | 1 | - | - | + | 1 | 2 | Step1: $z_1 = \sqrt{\frac{2}{3}} ln \frac{clay}{\sqrt{sand \times silt}}$ |
|      | 2 | + | - | 0 | 1 | 1 | Step2: $z_2 = \sqrt{\frac{1}{2}} ln \frac{sand}{silt}$ |


**2.4 Linear model, machine-learning model, and hybrid patterns**

**2.4.1 GLM**

The GLM is an extended version of the linear model, which contains response variables, with non-normal distributions (Nelder and Wedderburn, 1972). The link function is embedded into the GLM to ensure the classical linear model assumptions. The scaled dependent variables and the independent variables can be connected using a link function for the additive combination of model effects, the choice of link function depends on the distribution of response variables (Venables and Dichmont, 2004). A Gaussian distribution with an identity link function was applied in our study, which produced consequences equivalent to that of MLR (Nickel et al., 2014). However, categorical variables can be directly trained in the GLM without setting dummy variables. The Akaike's information criterion (AIC) was applied to choose the best predictors and remove model multicollinearity using a backward stepwise algorithm, and the combinations of ECs for different ILR data were then obtained (Table S2.1).

**2.4.2 RF**

The RF is a non-parametric technique, which combines the bagging method with a selection of random variables as an extended version of a regression tree (RT) (Breiman, 1996, 2001). It can improve model prediction accuracy by producing and aggregating multiple tree models. The principle of the RF is to merge a group of "weak trees" together to generate a "powerful forest." The bootstrap sampling method was applied for each tree, and each predictor was selected randomly from all model predictors. The "out of bag" (OOB) data were applied to produce reliable estimates in an internal validation using a random subset independent of the training tree data. Three parameters needed to be tuned: number of trees ($ntree$); minimum size of terminal nodes ($nodesize$), and number of variables randomly sampled as predictors for each tree ($mtry$) (Liaw and Wiener, 2001). The standard value of the $mtry$ parameter was one-third of the total number of predictors, while $ntree$ and





*nodesize* were 500 and 5, respectively. For regression, the mean square errors (MSEs) of predictions were estimated to train the trees. The variable importance of the RF was produced from the OOB data using the "importance" function. One of the benefits of the RF is that the ensembles of trees are used without pruning to ensure that the most significant amount of variance can be expressed. Moreover, the RF can reduce model overfitting and normalization is unnecessary due to the effects on the value range being insensitive. The GLM and RF algorithms and the parameter adjustment of the RF were conducted in the R package "caret" (Max Kuhn, 2018).

**2.4.3 RK**

Regression kriging is a hybrid interpolation technique that combines regression models (e.g., GLM and RF) with the residuals of OK (Odeh et al., 1995). Mathematically, the RK method corresponds to two interpolators, the regression part and the kriging part, which are operated separately (Goovaerts, 1999). One limitation of using only the regression part is that it is usually only useful within the range of values of the training sets (Hengl et al., 2015). The principle of the RK method is that the regression model explains a deterministic component of spatial variability, and the interpolation of regression residuals generated from OK is used to describe the spatial variability (Bishop and McBratney, 2001; Hengl et al., 2004). The residuals create a variogram (e.g., Gaussian, spherical, or exponential) for models based on the MSE from the results of a cross-validation. First, we used the regression part (GLM or RF) to predict soil PSFs, the residual from the fitted model was then calculated by subtracting the regression part from the observations. Subsequently, OK was applied for the whole study area to interpolate the residuals. Finally, the regression prediction and the predicted residuals at the same location were summed. The variograms of the RK method were generated automatically using the "autofitVariogram" function in the R package "automap" (Hiemstra et al., 2009).

**2.5 Prediction method system and validation**

The method system of spatial interpolation models for soil PSFs is presented in Table 2. We systematically compared 12 models: four interpolators, including GLM and RF with or without RK, and three SBPs of the ILR transformation method. For the validation of model performance, the independent data set validation was used to evaluate the prediction bias and accuracy of the models. The sub-training sets (70%) and the sub-testing sets (30%) were randomly selected from data independently, and this process was repeated 30 times.

**Table 2.** The method system of spatial interpolation models of soil PSFs.

| Models | GLM | GLMRK | RF | RFRK |
|---|---|---|---|---|
| ILR_SBP1 | GLM_SBP1 | GLMRK_SBP1 | RF_SBP1 | RFRK_SBP1 |
| ILR_SBP2 | GLM_SBP2 | GLMRK_SBP2 | RF_SBP2 | RFRK_SBP2 |
| ILR_SBP3 | GLM_SBP3 | GLMRK_SBP3 | RF_SBP3 | RFRK_SBP3 |





The mean error (ME), the root mean square error (RMSE), and Aitchison distance (AD) were used to evaluate and compare
the prediction performance. The ME and RMSE measure prediction bias and accuracy, respectively (Odeh et al., 1995). The
AD is an overall indicator of compositional analysis, which describes the distance between two compositions. Generally, in an
accurate, unbiased model all three values will be close to 0. The ME, RMSE, and AD were calculated as follows:
$ME = \frac{1}{n}\sum_{i=1}^{n}(M_i - P_i),$                                                                                                                                                       (7)
$RMSE = \sqrt{\frac{1}{n}\sum_{i=1}^{n}(M_i - P_i)^2},$                                                                                                                              (8)
$AD = \left[\sum_{i=1}^{D}(log\frac{M_i}{G(\boldsymbol{M})} - log\frac{P_i}{G(\boldsymbol{P})})^2\right]^{0.5},$                                                              (9)
where $M_i$ and $P_i$ are the measured and predicted values at the $i$th position, respectively; $n$ refers to the number of soil
samples; $D$ is the number of dimensions of compositions; and $G(\boldsymbol{M})$ and $G(\boldsymbol{P})$ denote the geometric mean with the form
$G(\mathbf{x}) = (x_1, \ldots, x_D)^{1/D}$ of the measured and predicted values, respectively.

**2.6 Covariance structure analysis**
The interpretation of the ILR balances is based on a decomposition of the covariance (COV) structure (Fiserova and Hron,
2011). We calculated the variance (VAR), COV, and the corresponding correlation coefficient (CC) of ILR transformed data
based on different SBP. The equations for calculating VAR, COV, and CC were derived from Eq. (1) as follows:
$VAR(z) = \frac{1}{r+s}\sum_{p=1}^{r}\sum_{q=1}^{s}var(ln\frac{x_{i_p}}{x_{j_q}}) - \frac{s}{2r(r+s)}\sum_{p=1}^{r}\sum_{q=1}^{r}var(ln\frac{x_{i_p}}{x_{i_q}}) - \frac{r}{2s(r+s)}\sum_{p=1}^{s}\sum_{q=1}^{s}var(ln\frac{x_{j_p}}{x_{j_q}}) -$
$\frac{r}{2s(r+s)}\sum_{p=1}^{s}\sum_{q=1}^{s}var(ln\frac{x_{j_p}}{x_{j_q}})$                                                                                                    (10)
$COV(z_1, z_2) = \frac{C}{2r_1s_2}\sum_{p=1}^{r_1}\sum_{q=1}^{s_2}var(ln\frac{x_{i_p^1}}{x_{j_q^2}}) + \frac{C}{2r_2s_1}\sum_{p=1}^{r_2}\sum_{q=1}^{s_1}var(ln\frac{x_{i_p^2}}{x_{j_q^1}}) - \frac{C}{2r_1r_2}\sum_{p=1}^{r_1}\sum_{q=1}^{r_2}var(ln\frac{x_{i_p^1}}{x_{i_q^2}}) -$
$\frac{C}{2s_1s_2}\sum_{p=1}^{s_1}\sum_{q=1}^{s_2}var(ln\frac{x_{j_p^1}}{x_{j_q^2}}),$                                                                                     (11)
$CC = \frac{COV(z_1,z_2)}{\sqrt{var(z_1)\cdot var(z_2)}}$                                                                                                                                    (12)
For soil PSF data, Eqs. (10), (11), and (12) can be simplified to three dimensions. The relationship between the ratios of soil
PSF components and the dominant roles of ILR transformed data were indicated from the covariance structure. All the
statistical analyses, such as the descriptive statistics of soil PSF data, calculation and evaluation of indicators, and the spatial
prediction mapping, were performed using the R statistical program (R Development Core Team, 2019).





## 3 Results

### 3.1 Exploratory data analysis

#### 3.1.1 Descriptive statistics of soil PSF data

From the descriptive statistics of the original (raw) and ILR transformed data, the silt fraction dominated the soil PSFs, accounting for a more substantial amount than the sand and clay fractions. The distributions of the sand and clay fractions were similar (Fig. 2a). The ILR transformed data based on the three SBPs revealed different distributions (Figs. 2b, 2c, and 2d). For example, two ILR components (ILR1 and ILR2) for SBP1 had a symmetric distribution around zero at the $x$-axis (Fig. 2b). In comparison, the distribution of data generated from SBP2 or SBP3 had a mirrored symmetry, with a left-skewed ILR1 of SBP2 and right-skewed ILR2 of SBP3 (Figs. 2c and 2d). The comparison of means and medians demonstrated that the back-transformed means of three sets of ILR transformed data were the same, and the mean ILR of sand was closer to the median compared with the original soil PSF data. In contrast, the opposite patterns were apparent for the silt and clay components (Fig. 2e).

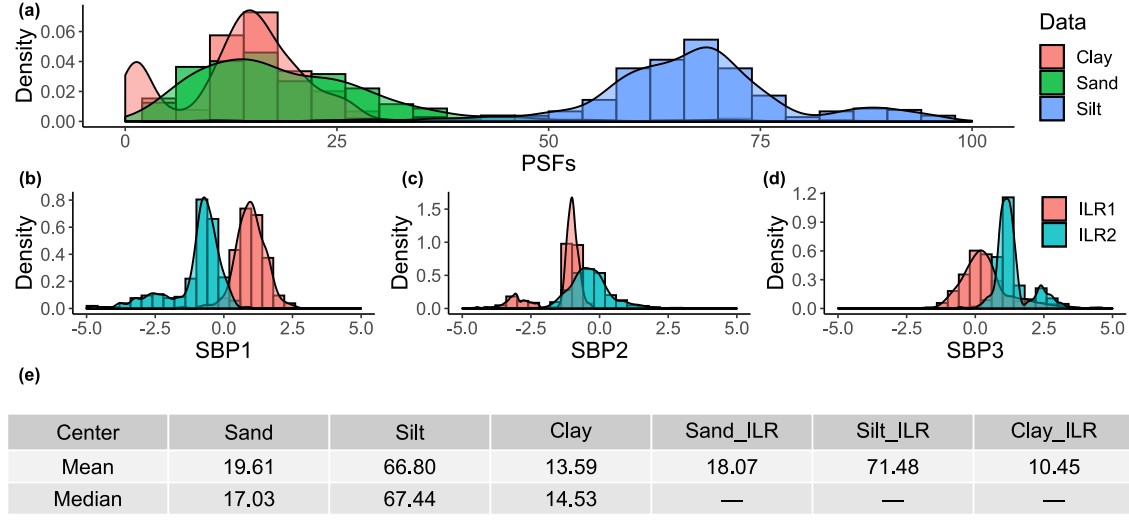

| Center | Sand | Silt | Clay | Sand_ILR | Silt_ILR | Clay_ILR |
|--------|------|------|------|----------|----------|----------|
| Mean | 19.61 | 66.80 | 13.59 | 18.07 | 71.48 | 10.45 |
| Median | 17.03 | 67.44 | 14.53 | — | — | — |

**Figure 2.** Descriptive statistics of original soil PSF and ILR transformed data using different SBPs. Note that means of Sand_ILR, Silt_ILR, and Clay_ILR from different SBPs were back-transformed to the real space.

### 3.1.2 Covariance structure of ILR transformed data with different balances

The covariance analysis of the transformed data of soil PSFs based on the different SBPs showed that the variance VarILR_1 of SBP3 was the largest, followed by the VarILR_1 of SBP1 and SBP2 (Table 3). The variance of the second component of ILR (VarILR_2) followed the opposite pattern to that of VarILR_1. The COV and corresponding CC followed the same pattern of SBP1 > SBP3 > SBP2. The first ILR equation ($z_1$ in Table 1) contained all information of soil PSFs, while the second one



($z_2$ in Table 1) included only two components. The information of VarILR_1 was therefore more abundant. All of VarILR_1
and VarILR_2 values were not 0 (or not nearly 0), indicating that there was no constant (or almost constant) value in any two
ratios of soil PSF components. The COV of SBP3 was close to 0, indicating that the proportions of *clay/sand* and *clay/silt*
were approximately the same. The same results were generated from the corresponding CC. For the distribution of soil PSFs
in a ternary diagram (the United States Department of Agriculture texture triangle, USDA), the main texture class was silt
loam (Fig. 3a). The biplot of soil samples demonstrated that the rays of the three components, i.e., sand, silt, and clay, were
reasonably well clustered at about 120° in the three groups (Fig. 3b).

**Table 3.** Covariance structure of soil PSFs based on different SBPs. VarILR_1 and VarILR_2 denote the variance of the first
and the second component of ILR, respectively. COV refers to the covariance of ILR1 and ILR2. CC is the correlation
coefficient.

| Balances | VarILR_1 | VarILR_2 | COV | CC |
|---|---|---|---|---|
| SBP1 | 0.53 | 0.71 | 0.32 | 0.52 |
| SBP2 | 0.39 | 0.86 | -0.24 | -0.41 |
| SBP3 | 0.94 | 0.30 | -0.09 | -0.16 |


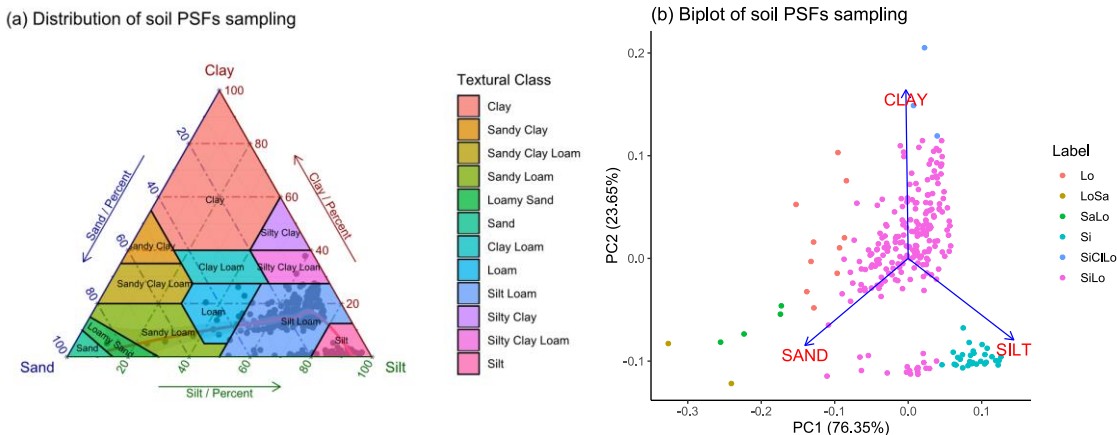


**Figure 3.** The distribution of soil PSFs in the USDA triangle (a) and biplot graph (b). The red curve was fitted by loess function.
**3.2 Accuracy comparison of different models using ILR data**
The first three rows of the boxplots in Figs. 4a, 4b, and 4c indicate the bias of the different models according to their ME
values. The ME of sand was closest to 0, followed by the MEs of clay and silt. GLM was more unbiased than RF, with lower
ME values. After combining with RK, there was an improvement in the ME for most GLM and RF models (Figs. 4a, 4b, and
4c). For the accuracy assessment, the RMSE of silt was higher than for the other two components. The GLMRK did not
perform as well as expected in terms of the RMSE, with only the sand component having an improved RMSE (Fig. 4d).
However, the RFRK performed better than the GLMRK and improved the accuracy of most parts compared with the RF,
except for the RFRK_SBP1 of sand. As an overall indicator, AD showed that the RF (or RFRK) performed better than the
GLM (or GLMRK) in terms of both average RMSE values and uncertainties (Fig. 4g). Moreover, the RFRK improved the AD
values for the SBP2 and SBP3 methods. For the uncertainty assessment, the RF generated lower uncertainties than the GLM,
and the models combined with RK further reduced the uncertainties for most GLM and RF models.

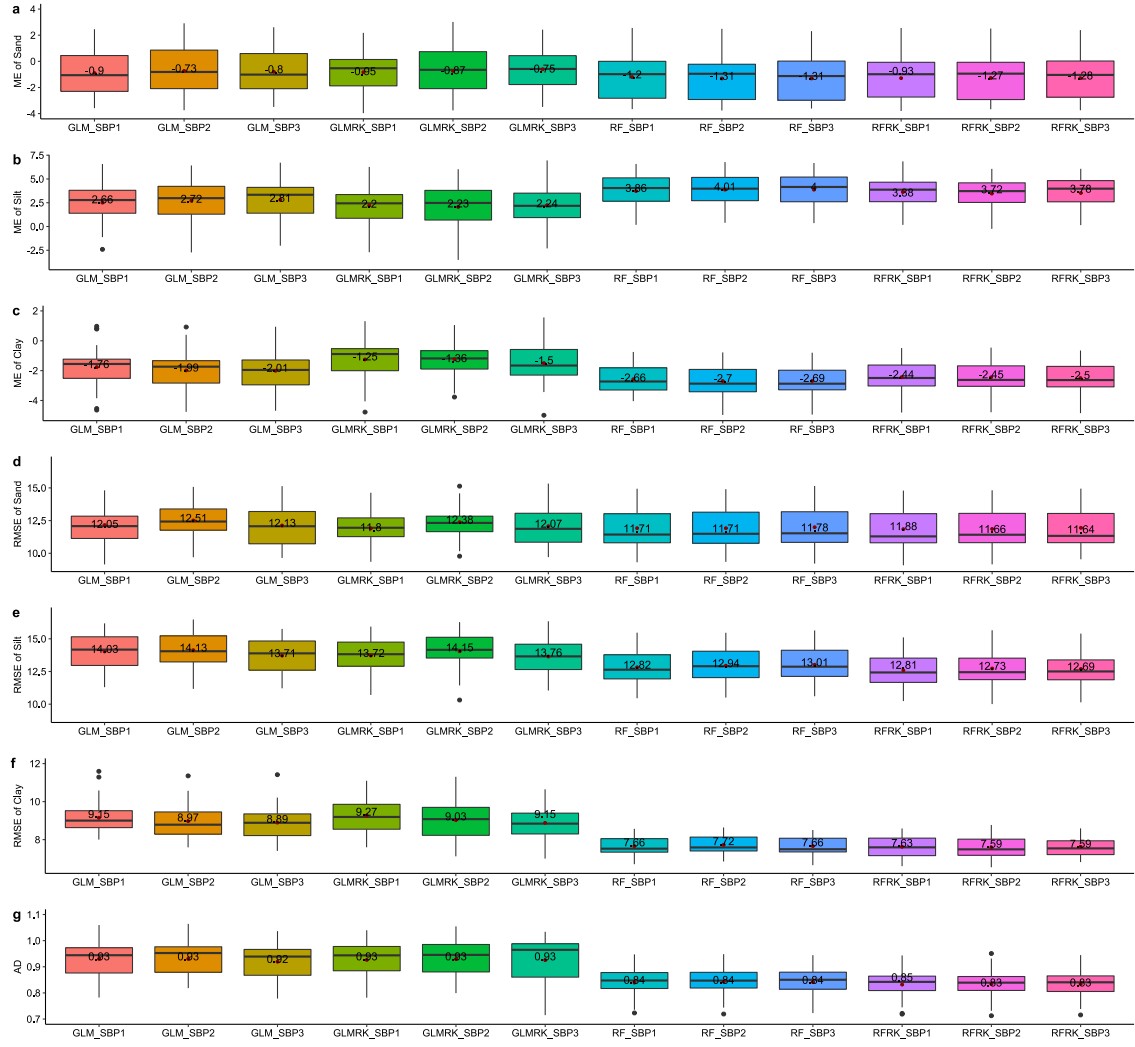


**Figure 4.** Accuracy comparison of GLM, RF, and their RK patterns combined with three ILR balances. The mean values of
different model indicators were calculated in their boxes.


The model performances were different for the three SBPs. To better evaluate model performance using the different SBP
balances, we graded each box from 1 to 3, and the final results are shown in Fig. 5. The results demonstrated that SBP1



performed best, with the lowest ME value of all models. For the accuracy comparison there was no apparent pattern, but
accuracy could be considered hierarchically: (1) for the GLM, SBP1 performed better than the other two SBP methods, which
also performed well when RK was combined (GLMRK); (2) for RF, SBP1 produced the best result. However, the introduction
of RK resulted in the Score2 of SBP3 performing best among the three SBPs. However, RFRK of SBP1 performed worst
according to the values of Score2 and Score5. Finally, for the comprehensive assessment, SBP1 performed best among three
SBPs according to Score6. More details and calculation processes can be found in the Supplementary Material (Table S4.1).

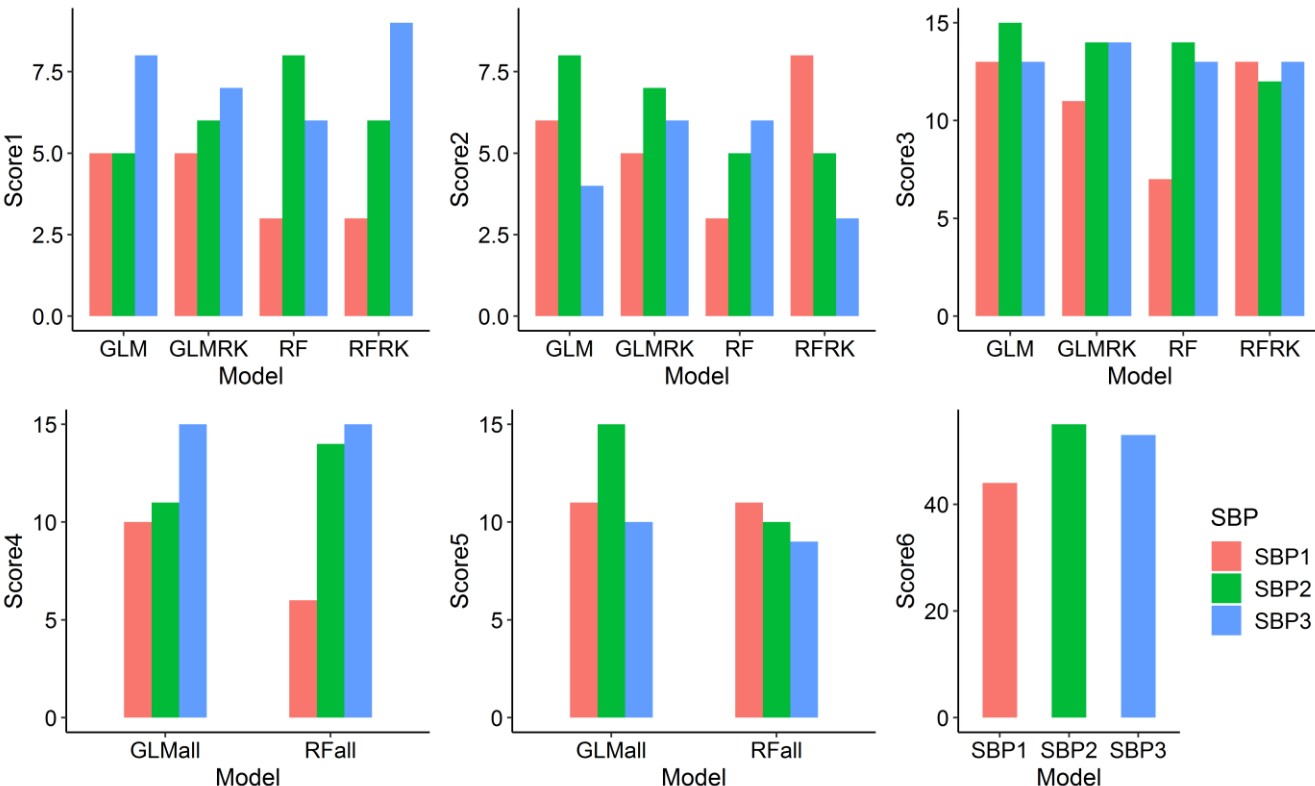


**Figure 5.** Ranking score of model performance based on three SBPs. Score1 and Score2 are the sum scores of ME and RMSE
for each model, respectively; Score3 is the sum scores of ME, RMSE and AD for each model, Score4 and Score5 are the sum
scores of ME or RMSE for GLM$_{all}$ (GLM and GLMRK) and RF$_{all}$ (RF and RFRK), Score6 is the sum scores of all indicators.
The lower the value, the better the model performance.

**3.3 Spatial prediction maps of soil PSFs generated from the different models**
Prediction maps of soil PSFs made from the different models are shown in Figs. 6, S3.1, and S3.2. For the components of soil
PSFs, the maps of the three group maps followed a similar rule. The GLM and GLMRK produced more extensive ranges of
predicted values, and their maps were more relevant to the real environment. However, the RF and RFRK predicted a relatively





narrow range of low values for these components, revealing a smoother distribution than that generated by the GLM and
GLMRK. Unlike the regression methods, the RF and RFRK methods produced hot and cold spots on the prediction maps and
more details of the soil sampling points were apparent (Fig. S5.1).

**(a) GLM_SBP1**     **(b) GLM_SBP2**     **(c) GLM_SBP3**

**(d) GLMRK_SBP1**     **(e) GLMRK_SBP2**     **(f) GLMRK_SBP3**

**(g) RF_SBP1**     **(h) RF_SBP2**     **(i) RF_SBP3**

**(j) RFRK_SBP1**     **(k) RFRK_SBP2**     **(l) RFRK_SBP3**

Sand (%)

| 0 - 11 | 16 - 20 | 24 - 29 | 36 - 50 |
| 11 - 16 | 20 - 24 | 29 - 36 | 50 - 75 |

0   40   80   160   240   320 km


**Figure 6.** Spatial prediction maps of the sand component of the upper reaches of the Heihe River Basin.
**3.4 Spatial distribution of soil texture classes in the USDA triangles**
The predicted soil textures in the USDA texture triangles (Fig. 7) showed that most predictions fell within the range of observed





soil textures (Fig. 3a), and silt loam was the dominant soil texture in all cases. The GLM produced a more discrete distribution
than the RF, and the RK method expanded the dispersion. In the trends of the predicted samples, the silt components predicted
from all models were overestimated. The pattern fitting curves indicated that the prediction results were closer to the bottom
right of the USDA triangle than the soil PSF observations. The GLMRK and RFRK curves were longer than the GLM and RF
curves, with a more extensive range of values in triangles. Compared with the GLMRK, the RFRK produced a more upward
extension (Figs. 7j, k, l). It was clear that the clay fraction was overestimated and the sand fraction was underestimated.

**Figure 7.** Predicted 262 soil samples in USDA texture triangles using (a) GLM_SBP1, (b) GLM_SBP2, (c) GLM_SBP3, (d) GLMRK_SBP1, (e) GLMRK_SBP2, (f) GLMRK_SBP3, (g) RF_SBP1, (h) RF_SBP2, (i) RF_SBP3, (j) RFRK_SBP1, (k) RFRK_SBP2, (l) RFRK_SBP3. Red fitting curves in triangles showed the trends.



## 4 Discussion

### 4.1 Comparison of the GLM, RF, and RK patterns using ILR data

We found RF reveal more accurate results, but with more bias than the GLM, and RK method improved the performance in
terms of bias for most models and the accuracy of the RF. Odeh et al. (1995) indicated that RK was superior to the linear
models, such as MLR, which was reflected in the prediction results for sand in our study. Scarpone et al. (2016) reported that
as a hybrid interpolator, the RFRK outperformed the RF when making soil thickness predictions. We proved that RFRK was
also suitable for compositional data and improved model performance when combining with the ILR transformation. In
summary, the GLM and RF had both advantages and disadvantages when considering the trade-off between bias and accuracy.
The results of GLM and GLMRK should not depend on the ILR basis being chosen, which has been proved by previous
studies on the use of linear models and kriging for compositional data (Pawlowsky-Glahn et al, 2015). However, the GLM
model used the "glmStepAIC" algorithm (i.e., a stepwise regression) to select the best combination of environmental
covariables for each ILR component (Table S2.1). Therefore, the variable inputs are different for these ILR data, and further
impact the accuracy assessment and prediction maps. In addition, the difficulty with the use of the GLM is the need for a back-
transformation. There is a need to present results on the original untransformed scale after conducting the analysis on a
transformed level, which may produce spurious results (Lane, 2002). In our study, we compared the means of ILR transformed
data and the original data. We proved the feasibility of the ILR transformation method, especially for meeting the requirements
of compositional data. However, the accuracy of the GLM still needs to be improved, which may be because the transformed
data did not follow a normal distribution (Fig. 2).
Although the RF had the advantage of prediction accuracy, the limited interpretability of the consequences made it difficult
to modify the prediction bias – each tree from the model cannot be examined individually (Grimm et al., 2008). Moreover, the
ILR transformation before modeling increased the difficulty of interpretation for not only the predicted values on the ILR scale
but also the residuals. The back-transformation of the optimal estimate of log-ratio variables does not generate the optimal
estimation of compositional data (Lark and Bishop, 2007), which should also be considered.

### 4.2 Comparison of three SBPs of ILR transformation

For the comparison of the three SBPs, the ME and RMSE performed better when using SBP1 for ILR transformed data,
which may be interpreted as the distributions of the ILR1 and ILR2 of SBP1 being more symmetric (Fig. 2b). In contrast, the
performance of SBP2 was worse than that of SBP1 and SBP3 because the ILR_1 component, including all the soil PSF
information, was left-skewed (Fig. 2c). This result was especially apparent for the GLM and GLMRK, because the data in a
linear model needs to be normally distributed (Lane, 2002).
The negligible difference among the three SBP balances revealed a triangular shape with a cluster at about 120° (Fig. 3b).
This could be interpreted as the three soil PSFs having a mixed pattern, with each component dominated by the components
in one cluster (Tolosana-Delgado et al., 2005). Although the silt component dominated the soil PSFs (Fig. 2a), sand and clay



also played important roles in soil compositions. Taking either the most abundant component of the compositional data as the
denominator (Martins et al., 2016) or the first component of the permutations did not provide convincing evidence. Because
using the most abundant component of the compositional data as the primary component of the alterations, i.e., SBP2, resulted
in a relatively poor performance compared to the other SBPs. Thus, we recommend that the focus should be on data distribution.
Furthermore, the choice of balance and combination of RK are also the key to improving model accuracy, as shown by the
result of the RFRK-SBP3 model (Fig. 4).
**4.3 Limitations**
Firstly, the scope of this study is limited to independent modeling. Each ILR component was modeled separately, which may
suboptimal because they cannot further consider the cross correlations among ILR coordinates. However, the study
demonstrated the relation of the raw data (sand, silt, and clay), and has confirmed that the currently used prediction models are
suitable. In our pervious study, we have used compositional kriging (CK) for the spatial prediction of soil PSFs (Wang and Shi,
2017), and the cross correlations of ILRs can be taken into account using CK. Although it is optimal, it cannot consider different
balances of ILR, nor can it be combined with the hybrid interpolator (e.g., RK). Moreover, predicting each ILR component
separately was a more suitable approach for the spatial prediction models currently used (such as the GLM and RF). Therefore,
more alternative spatial prediction models combined with interpretation of ILR balances for compositional data should be
considered in the future. For example, CK and high accuracy surface modelling (HASM; Yue et al., 2016) can be applied for
small scale study areas. For large scale study areas, multivariate RF (Segal and Xiao, 2011) can be combined with a log-ratio
transformation and hybrid interpolation method, enabling the cross correlations among ILR coordinates to be better interpreted.
Secondly, the weighting problem was not considered in this study, because the ILR method can be qualified as an unweighted
log-ratio transformation, giving all parts the same weight for both the definition of the total variance and the reduction of
dimension. This may enlarge the ratios generated from the rare parts, which would dominate the analysis (Greenacre and Lewi,
2009). The pairwise log-ratio can be used to set weights by their proportions when there is no additional knowledge about the
component measurement errors (Greenacre, 2019). Nevertheless, all three parts of the soil PSF data dominated the biplot
diagram, without the influence of rare elements and with no redundancy; thus, none of the shortcomings mentioned above
were apparent. Accuracy assessments using a pairwise log-ratio transformation require further study in the future.
**5 Conclusions**
We evaluated and compared the performance of the GLM, RF, and their hybrid pattern (i.e., GLMRK and RFRK) using
different balances of ILR transformed data. The bias of the GLM was lower than that of the RF; however, the accuracy of the
GLM was relatively low. More discrete distributions and broader ranges of prediction value distributions were produced from
GLMs in the USDA soil texture triangles. In other words, different predicted data sets were generated from the use of the GLM
and RF, with unbiased and inaccurate predictions for the GLM and biased and more accurate predictions for the RF.
The hybrid patterns, GLMRK and RFRK, were found to be the best solution because it produced a relatively high prediction
accuracy and strong correlations with ECs, providing more details about the soil sampling points (hot spots and cold spots)





compared with only the regression model. However, the non-normal distribution of ILR data and its residuals, and more data
transformation and inverse transformation processes make models further difficult to interpreted and improve.
For the different SBPs, the three SBP-based data generated different distributions, but no pattern was apparent. This could
be explained by the angle of the biplot diagram, with three rays of soil PSF components clustered into three modes, and each
part dominating its cluster. Using the most abundant component of the compositional data as the first component of the
permutations was not considered the right choice because SBP2 produced the worst performance. Thus, we recommend that
the focus should be on data distribution. This study can provide a reference for the spatial simulation of soil PSFs combined
with ECs at the regional scale, and how to choose the balances of ILR transformed data.

***Data Availability.*** We did not use any new data and the data we used come from previously published sources. Soil particle-
size fractions data is available through our previous studies (Wang and Shi, 2017, 2018). Moreover, it also can be visited on
this website: http://data.tpdc.ac.cn/zh-hans/data/7f91d36d-8bbd-40d5-8eaf-7c035e742f40/ (Digital soil mapping dataset of
soil texture (soil particle-size fractions) in the upstream of the Heihe river basin (2012-2016); last access: 4 July 2020). The
meteorological data can be accessed through http://data.cma.cn/ (last access: 4 July 2020). Environmental covariates data of
soil physical and chemical properties and categorical maps can be obtained through http://data.tpdc.ac.cn/zh-hans/ (last access:
4 July 2020), including saturated water content, field water holding capacity, wilt water content, saturated hydraulic
conductivity data (http://data.tpdc.ac.cn/zh-hans/data/e977f5e8-972b-42a5-bffe-cd0195f3b42b/, Digital soil mapping dataset
of hydrological parameters in the Heihe River Basin (2012); last access: 4 July 2020), and soil thickness data
(http://data.tpdc.ac.cn/zh-hans/data/fc84083e-8c66-4a42-b729-4f19334d0d67/, Digital soil mapping dataset of soil depth in
the Heihe River Basin (2012-2014); last access: 4 July 2020). DEM data set is provided by the Geospatial Data Cloud site,
Computer Network Information Center, Chinese Academy of Sciences. (http://www.gscloud.cn, last access: 4 July 2020).

***Author contribution.*** Wenjiao Shi contributed to soil data sampling, oversaw the design of the entire project. Mo Zhang
performed the model analysis and wrote the manuscript. Both authors contributed to writing this paper and interpreting data.

***Competing interests.*** The authors declare that they have no conflict of interest.

***Acknowledgment.*** Our team expresses gratitude to the following institutions, Key Laboratory of Land Surface Pattern and
Simulation, Institute of Geographic Sciences and Natural Resources Research, Chinese Academy of Sciences; School of Earth
Sciences and Resources, China University of Geosciences; College of Resources and Environment, University of Chinese
Academy of Sciences. This study was supported by the National Key Research and Development Program of China (No.
2017YFA0604703), the National Natural Science Foundation of China (Grant No. 41771111 and 41771364), Fund for
Excellent Young Talents in Institute of Geographic Sciences and Natural Resources Research, Chinese Academy of Sciences
(2016RC201), and the Youth Innovation Promotion Association, CAS (No. 2018071).



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
