# Peer review of "Compositional balance should be considered in the mapping of soil particle-size fractions using hybrid interpolators"

_Hydrology and Earth System Sciences, 2021_

## Author Comment (AC1)

**Responses to the Referee**

**Dear authors please see the following comments: I think the major problem is the presentation of an extended uncertainty analysis.**

**Comment 1: why 29 ECs and not groups of categorical and continuous or mixture of them, a test is needed like PCA and others?**

**Response:** Thank you for your suggestion of including the combinations and PCA test of ECs. The 29 ECs we used in this work were 29 the results of removing highly correlated variables. We selected 45 ECs initially. Considering that models may benefit from reducing the level of correlation between the variables, absolute correlations above 0.7 were removed using correlation analysis and the "findCorrelation" function in the 'caret' R package. In addition, for the combinations of different models, we applied the backward stepwise algorithm and obtained different combinations of ECs for different ILR data.

**P4L252**: *"There were 29 ECs considered in our study after reducing the level of correlation between the variables, including both continuous and categorical variables (Table S1.1)."*

**P6L299**: *"The Akaike's information criterion (AIC) was applied to choose the best predictors and remove model multicollinearity using a backward stepwise algorithm, and the combinations of ECs for different ILR data were then obtained (Table S2.1)."*

**Comment 2: It is not clear mathematically how you apply the proposed methodologies.**

**Response:** Thank you for your suggestion regarding the proposed methodologies. The simplified formula and flow chart are as follows:

For the simplified formula, when the ILR method was applied in soil PSFs (D=3), three components (i.e., sand, silt, and clay) were transformed into two components (i.e., ILR1 and ILR2). Moreover, using different SBPs (in total, three types of SBPs), we applied different permutations of three components to derive the final formulas for three SBPs (Table 1). The transformation process is available in the 'compositions' R package using the "ilr" function.

The flow chart showed how ILR transformed data were applied to the RK model. We used ILR transformed data (ILR1 and ILR2) to predict models and their residuals; then the two parts (predicted ILR1 and ILR2) were added and back-transformed into predicted soil PSF data (sand, silt and clay).

**Table 1.** *All choices of SBPs for soil PSF data (D = 3), the order of soil PSFs data is* $(sand, silt, clay)$, $(silt, clay, sand)$, *and* $(clay, sand, silt)$ *for SBP1, SBP2 and SBP3.*

| Groups | Step | Sand | Silt | Clay | r | s | Formula |
|--------|------|------|------|------|---|---|---------|
| SBP1 | 1 | + | - | - | 1 | 2 | Step1: $ILR1 = \sqrt{\frac{2}{3}} \ln \frac{sand}{\sqrt{silt \times clay}}$ |
| | 2 | 0 | + | - | 1 | 1 | Step2: $ILR2 = \sqrt{\frac{1}{2}} \ln \frac{silt}{clay}$ |
| SBP2 | 1 | - | + | - | 1 | 2 | Step1: $ILR1 = \sqrt{\frac{2}{3}} \ln \frac{silt}{\sqrt{clay \times sand}}$ |
| | 2 | - | 0 | + | 1 | 1 | Step2: $ILR2 = \sqrt{\frac{1}{2}} \ln \frac{clay}{sand}$ |
| SBP3 | 1 | - | - | + | 1 | 2 | Step1: $ILR1 = \sqrt{\frac{2}{3}} \ln \frac{clay}{\sqrt{sand \times silt}}$ |
| | 2 | + | - | 0 | 1 | 1 | Step2: $ILR2 = \sqrt{\frac{1}{2}} \ln \frac{sand}{silt}$ |

[Figure]

**Fig. 1.** *Process of RK method in our study.*

**Comment 3: For RK: you should provide more details about the RK process: regression type, variogram types, parameters, nugget, fitting method, suitability of data for geostatistical analysis etc.**

**Response:** Thank you for your suggestion regarding the details of the RK process. We have updated the details of the RK process in the Supplementary Material, which now includes two figures–RK of GLM and RF (i.e., GLMRK and RFRK), respectively. Variogram types, parameters are included.

[Figure]

***Figure S6.1.*** *Variograms of GLM using different ILR transformed data.*

[Figure]

***Figure S6.2.*** *Variograms of RF using different ILR transformed data.*

**Comment 4: Table 2 needs better explanation.**

**Response:** Thank you for your suggestion. Table 2 showed all the models we built, which were combinations of four models (GLM, GLMRK, RF, RFRK) and three types of SBP. Therefore, a total of 12 models were predicted, and their performance compared.

**P8L332** *"The method of spatial interpolation for soil PSFs is presented in Table 2. We systematically compared 12 models: the combinations of four interpolators (GLM, GLMRK, RF, RFRK), and three SBPs of the ILR transformation method."*

***Table 2.*** The method of spatial interpolation of soil PSFs.

| Models | GLM | GLMRK | RF | RFRK |
|---|---|---|---|---|
| SBP1 | GLM_SBP1 | GLMRK_SBP1 | RF_SBP1 | RFRK_SBP1 |
| SBP2 | GLM_SBP2 | GLMRK_SBP2 | RF_SBP2 | RFRK_SBP2 |
| SBP3 | GLM_SBP3 | GLMRK_SBP3 | RF_SBP3 | RFRK_SBP3 |

**Comment 5: It is not explained how the uncertainty has been calculated. A more clear and extended presentation and calculation of uncertainty is required.**

**Response:** Thank you for your suggestion regarding the calculation of uncertainty. We compared the uncertainties by calculating the ranges of 95% confidence interval (CI) (Streiner, 1996) derived from running models 30 times (i.e., CI of ME, RMSE, and AD). The box diagrams also showed the uncertainties of the different models. We have added calculation methods in our revised manuscript.

**P8L282:** *"The ranges of a 95% confidence interval (CI) (Streiner, 1996) of ME, RMSE, and AD were calculated to compare the uncertainties of different models."*

**Comment 6: Generally different algorithms have been applied but it is unclear how the uncertainty propagation affects the final results.**

**Response:** The results of uncertainty analysis for different models were demonstrated in the added table in the Supplementary Material (Table. S7.1). The main reasons why uncertainty propagation affects the final results are: (1) input data are different based on three SBPs, and (2) models we applied are different. Firstly, the input data of ILR methods were different (three SBPs), and these different ILR data directly impacted the prediction results and uncertainty. Different input data generate different SBPs of ILR, which means we should consider the SBP in soil PSF interpolation. Secondly, the main differences in these applied models were linear regression (GLM) and machine-learning method (RF), and models with or without RK. The results showed that CI_ME of GLM were lower than that of RF, but CI_RMSE and CI_AD of RF delivered a better performance. Moreover, introducing of RK can reduce the uncertainty, especially for the sand fraction. For the uncertainty of prediction maps, we added the ordinary kriging variance and the range of 95% prediction interval of different models in the
Supplementary Material. Because of the small values of variance for the ILR data, the differences of interval were very close,
showing low uncertainty when using ILR transformed data, which was also an advantage that can be considered.

**P18L488**: *"With respect to uncertainty, the uncertainty of bias for GLM is higher than that of RF, but the uncertainty of*
*accuracy for GLM is lower. However, RF performed better in terms of accuracy assessment. Therefore, the main concern was*
*whether the introductions of RK could reduce the uncertainty of RF. With regard to the performances of RFRK and RF, adding*
*RK was recommended in soil PSF interpolation combined with ILR transformed data. In addition, the range of 95% prediction*
*interval for different models (Figs. S8.1–8.6) demonstrated that the differences were very close. This may because the values*
*of variance for ILR data were small, showing low uncertainty when using ILR transformed data."*

***Table. S7.1.*** *The ranges of 95 % confidence interval (CI) of ME, RMSE and AD for different models.*

| | CI_ME | | | CI_RMSE | | | CI_AD |
|---|---|---|---|---|---|---|---|
| | sand | silt | clay | sand | silt | clay | |
| GLM_SBP1 | 1.20 | 1.65 | 1.02 | 1.16 | 0.94 | 0.63 | 0.04 |
| GLM_SBP2 | 1.39 | 1.74 | 0.99 | 0.98 | 0.87 | 0.67 | 0.04 |
| GLM_SBP3 | 1.22 | 1.58 | 0.95 | 1.15 | 0.96 | 0.62 | 0.05 |
| GLMRK_SBP1 | 1.16 | 1.56 | 1.03 | 1.04 | 0.88 | 0.69 | 0.05 |
| GLMRK_SBP2 | 1.38 | 1.75 | 1.02 | 0.94 | 1.03 | 0.74 | 0.05 |
| GLMRK_SBP3 | 1.22 | 1.56 | 0.97 | 1.08 | 1.03 | 0.95 | 0.05 |
| RF_SBP1 | 1.26 | 1.44 | 0.69 | 1.26 | 0.97 | 0.38 | 0.04 |
| RF_SBP2 | 1.24 | 1.40 | 0.68 | 1.25 | 0.99 | 0.37 | 0.04 |
| RF_SBP3 | 1.26 | 1.43 | 0.67 | 1.30 | 1.02 | 0.37 | 0.04 |
| RFRK_SBP1 | 1.21 | 1.35 | 0.71 | 1.23 | 0.99 | 0.40 | 0.04 |
| RFRK_SBP2 | 1.19 | 1.32 | 0.69 | 1.21 | 1.03 | 0.39 | 0.04 |
| RFRK_SBP3 | 1.20 | 1.32 | 0.69 | 1.25 | 1.01 | 0.39 | 0.04 |

[Figure]

*Figure 1.* Ordinary kriging variance for GLM using different SBPs.

[Figure]

**Figure 2.** *Ordinary kriging variance for RF using different SBPs.*

VIII

[Figure]

*'Supplementary Material Section S8. The uncertainty of prediction maps'*

**Figure S8.1.** *95% prediction intervals of soil PSFs using GLM combined with SBP1.*

[Figure]

**Figure S8.2.** *95% prediction intervals of soil PSFs using GLM combined with SBP2.*

[Figure]

***Figure S8.3.*** *95% prediction intervals of soil PSFs using GLM combined with SBP3.*

[Figure]

***Figure S8.4.*** *95% prediction intervals of soil PSFs using RF combined with SBP1.*

[Figure]

**119**

**120**     *Figure S8.5.* *95% prediction intervals of soil PSFs using RF combined with SBP2.*

**121**

[Figure]

**122**

**123**     *Figure S8.6.* *95% prediction intervals of soil PSFs using RF combined with SBP3.*

**124**

**125**     **Comment 7: The mixture of categorical and continues data needs more explanation in terms of methods applications**

**126**     **and uncertainty of the results.**

**Response:** We added this discussion in the Discussion Section. Firstly, for categorical and continuous data, the processes of the linear model and machine-learning model were different. GLM applied a backward stepwise algorithm and processed categorical data as dummy variables. Thus, not all ECs we selected were used in the process of model training for GLM (Table S2.1). In addition, RF applied all variables and did not require data processing for categorical variables. For the uncertainty in the results, the uncertainty in bias for GLM was higher than that of RF, but the uncertainty in accuracy for GLM was lower. However, RF performed better in the accuracy assessment; the main concern, therefore, was whether introducing RK could reduce the uncertainty in RF. The results showed that the uncertainty in accuracy for RFRK was lower than that of RF. Thus, RFRK was recommended in soil PSF interpolation combined with ILR transformed data.

**P18L481:** *"The results of GLM and GLMRK should not depend on the choice of ILR basis, which has been proved by previous studies on the use of linear models and kriging for compositional data (Pawlowsky-Glahn et al, 2015). However, the GLM model used the "glmStepAIC" algorithm (i.e., a stepwise regression) to select the best combination of environmental covariables for each ILR component (Table S2.1). Therefore, the variable inputs are different for these ILR data, and further impact the accuracy assessment and prediction maps."*

**P18L488:** *"With respect to uncertainty, the uncertainty of bias for GLM is higher than that of RF, but the uncertainty of accuracy for GLM is lower. However, RF performed better in terms of accuracy assessment. Therefore, the main concern was whether the introductions of RK could reduce the uncertainty of RF. With regard to the performances of RFRK and RF, adding RK was recommended in soil PSF interpolation combined with ILR transformed data. In addition, the range of 95% prediction interval for different models (Figs. S8.1–8.6) demonstrated that the differences were very close. This may because the values of variance for ILR data were small, showing low uncertainty when using ILR transformed data."*

**Compositional balance should be considered in the mapping of soil particle-size fractions using hybrid interpolators**

Mo Zhang[1,2], Wenjiao Shi[1,2]

[1]Key Laboratory of Land Surface Pattern and Simulation, State Key Laboratory of Resources and Environmental Information System, Institute of Geographic Sciences and Natural Resources Research, Chinese Academy of Sciences, Beijing 100101, China

[2]College of Resources and Environment, University of Chinese Academy of Sciences, Beijing 100049, China

*Correspondence to:* Wenjiao Shi (shiwj@lreis.ac.cn), Institute of Geographic Sciences and Natural Resources Research, Chinese Academy of Sciences. 11A, Datun Road, Chaoyang District, Beijing 100101, China.

**Abstract**. Digital soil mapping of soil particle-size fractions (PSFs) using log-ratio methods is a widely used technique. As a hybrid interpolator, regression kriging (RK) is one way to improve prediction accuracy of soil PSFs. However, there is still a lack of comparisons and recommendations when RK is applied for compositional data. It is unknown if the prediction performance based on different balances of the isometric log-ratio (ILR) transformation is robust. We compared the generalized linear model (GLM), the random forest (RF) model, and their hybrid patterns (i.e., GLMRK and RFRK) using different transformed data based on three ILR balances. The comparison involved 29 environmental covariables  for the  soil PSF prediction in the upper reaches of the Heihe River Basin, China. The results showed that RF performed best, with more accurate predictions, but GLM produced a more unbiased prediction. For the hybrid interpolators, RK was recommended because it widened the data ranges of the prediction values and modified the bias and accuracy for most models, especially for RF. Moreover, prediction maps generated from RK revealed more details of the soil sampling points. For three ILR balances, different data distributions were produced. Using the most abundant component of the compositional data as the first component of the permutations was not considered the best choice for soil PSF mapping. Compared to the relative abundance of components, we recommend that the focus should be on data distribution. This study provides a reference for  mapping  soil PSFs combined with transformed data at the regional scale.

**1 Introduction**

Recently, spatial interpolation of soil particle-size fractions (PSFs) has become a focus of soil science researchers. More accurately predicted soil PSFs elucidates hydrological, physical, and environmental processes (Delbari et al., 2011; Ließ et al., 2012; McBratney et al., 2002).

The characteristics of compositional data make soil PSFs more impressive than other soil properties. Soil PSFs are usually expressed as three components for discrete data  sand, silt, and clay and carry only relevant percentage information. Soil texture is classified by soil PSFs, which can be demonstrated on a ternary diagram (so-called soil texture triangle). However, the closure system formed in this triangle is not in Euclidean space, but  rather Aitchison space (i.e., a simplex) (Aitchison, 1986). Due to "spurious correlations" (Pawlowsky-Glahn, 1984), traditional statistical methods based on  Euclidean geometry may generate mistakes when dealing directly with soil PSF data (Filzmoser et al., 2009). The requirement for constant-sum, nonnegative, unbiased prediction is the key to spatial interpolation (Walvoort and de Gruijter, 2001). Therefore, data transformation from a simplex into real space is crucial for compositional data . Log ratio transformations play a significant role in compositional data analysis, including the additive log-ratio (ALR), centered log-ratio (CLR) (Aitchison, 1986), and the isometric log-ratio (ILR) (Egozcue et al., 2003).

Although these three log-ratio methods have been widely applied to transform soil PSF data, different study area scales and model selection should be considered when modeling. For local-scale study areas, geostatistical models combined with log-ratio transformed data are sufficient to map spatial patterns, as shown in our previous study (Wang and Shi, 2017). From another perspective, functional compositions combined with the kriging method can also be applied to produce soil particle size curves (PSCs) (Menafoglio et al., 2014), providing an abundance of information. Functional compositions involve the use of complete and continuous information rather than discrete information, and soil PSFs can be extracted from the predicted soil PSCs (Menafoglio et al., 2016a). Log-ratio transformations can also be combined with functional-compositional data for the stochastic simulation of PSCs (Menafoglio et al., 2016b, Talska et al., 2018). For middle-scale study areas, outliers may lead to  overestimation of the variogram, resulting in prediction errors (Lark, 2000). Therefore,  spatial interpolation should take robust variogram estimators into account to improve model performance (Lark, 2003). A previous study proved that applying robust variogram estimators in log-ratio co-kriging significantly improved mapping performance (Wang and Shi, 2018). For large-scale study areas, geostatistical models are limited by the number of soil samples and increased spatial variability. Increasing numbers of studies have concentrated on mapping soil PSFs using different machine-learning models combined with ancillary data (i.e., environmental covariables, ECs). Log-ratio transformed data have been applied on a broad basin scale (Zhang et al., 2020), national scale (Akpa et al., 2014), and even a global scale (Hengl et al., 2017).

Among these EC-combined models, linear, machine-learning, geostatistical models, and high-accuracy surface modeling (Yue et al., 2020; Shi et al., 2016) have been commonly used in middle or large-scale studies. Linear models, for example, the generalized linear model (GLM) and multiple linear regression (MLR) model, have been used in soil PSF predictions because of their flexibility and interpretability (Lane, 2002; Buchanan et al., 2012). Many machine-learning models have been applied for  soil PSF interpolation and soil texture classification. For example, tree learners, such as the random forest (RF) model, are advantageous due to their ability to handle overfitting and generate more realistic maps (Zhang et al., 2020). In addition, regression kriging (RK) has proven to be a powerful and widely accepted method for soil interpolation. ECs can be introduced through its regression function and improved model accuracy as a hybrid interpolator for some soil properties, such as topsoil thickness and pH (Hengl et al., 2004; Keskin and Grunwald, 2018; Shi et al., 2009; Shi et al., 2011). However, the scope of the comparison needs to be expanded to further explore the prediction accuracy combined with compositional data using linear models, machine-learning models, and combining RK with other models  (hybrid patterns).

In log-ratio methods, the ILR method performs better than the ALR and CLR methods in both theory and  practice (Filzmoser and Hron, 2009; Wang and Shi, 2018; Zhang et al., 2020). The ILR method eliminates model collinearity and preserves advantageous properties, such as isometry, scale invariance, and sub-compositional coherence. The ILR method is constructed in orthonormal coordinate systems (i.e., balances) using a sequential binary partition (SBP) (Egozcue and Pawlowsky-Glahn, 2005). The choices of balances are not unique. Multiple sets of ILR transformed data can be generated by permutations of components (different SBPs) in the compositional data. The selection of an SBP can be based on prior expert knowledge, using a compositional biplot (Lloyd et al., 2012) or variograms and cross-variograms (Molayemat et al., 2018). It has been proven in statistical science that different results are obtained using different ILR balances. For example, Fiserova and Hron (2011) reported that different balances generated different covariance structures. Moreover, the choice of SBP is related to hypotheses, research questions of interest, or the context of the data analysis (Coenders et al., 2017; Facevicova et al., 2018). Thus, 
[revised manuscript text omitted]

**2.5 Prediction method system and validation**

The method  of spatial interpolation  for soil PSFs is presented in Table 2. We systematically compared 12 models: the combinations of four interpolators (GLM , GLMRK, RF , RFRK), and three SBPs of the ILR transformation method. For the validation of model performance,  independent data set validation was used to evaluate the prediction bias and accuracy of the models. The  data were randomly divided into two sets: the sub-training sets (70%) and the sub-testing sets (30%), and this process was repeated 30 times. Moreover, the Diebold–Mariano test (Diebold and Mariano, 1995; Harvey et al., 1997) was used to verify the statistical significance of the differences among the models.

**Table 2.** The method  of spatial interpolation  of soil PSFs.

| Models | GLM | GLMRK | RF | RFRK |
|---|---|---|---|---|
|  SBP1 | GLM_SBP1 | GLMRK_SBP1 | RF_SBP1 | RFRK_SBP1 |
|  SBP2 | GLM_SBP2 | GLMRK_SBP2 | RF_SBP2 | RFRK_SBP2 |
|  SBP3 | GLM_SBP3 | GLMRK_SBP3 | RF_SBP3 | RFRK_SBP3 |

The mean error (ME), the root mean square error (RMSE), and Aitchison distance (AD) were used to evaluate and compare the prediction performance. The ME and RMSE measure prediction bias and accuracy, respectively (Odeh et al., 1995). The AD is an overall indicator of compositional analysis, which describes the distance between two compositions. Generally, in an accurate, unbiased model all three values will be close to 0. The ranges of a 95% confidence interval (CI) (Streiner, 1996) of ME, RMSE, and AD were calculated to compare the uncertainties of different models. The ME, RMSE, and AD were calculated as follows:

$$ME = \frac{1}{n}\sum_{i=1}^{n}(M_i - P_i), \tag{7}$$

$$RMSE = \sqrt{\frac{1}{n}\sum_{i=1}^{n}(M_i - P_i)^2}, \tag{8}$$

$$AD = \left[\sum_{i=1}^{D}(log\frac{M_i}{G(M)} - log\frac{P_i}{G(P)})^2\right]^{0.5}, \tag{9}$$

where $M_i$ and $P_i$ are the measured and predicted values of the $i$th sample for sand, silt and clay; $n$ refers to the number of soil samples; $D$ is the number of dimensions of compositions; and $G(\boldsymbol{M})$ and $G(\boldsymbol{P})$ denote the geometric mean with the form $G(\mathbf{x}) = (x_1, \ldots, x_D)^{1/D}$ of the measured and predicted values, respectively.

**2.6 Covariance structure analysis**

The interpretation of the ILR balances is based on a decomposition of the covariance (COV) structure (Fiserova and Hron, 2011). We calculated the variance (VAR), COV, and the corresponding correlation coefficient (CC) of ILR transformed data based on different SBPs. The equations for calculating VAR, COV, and CC were derived from Eq. (1) as follows:

$$VAR(z) = \frac{1}{r+s} \sum_{p=1}^{r} \sum_{q=1}^{s} var\left(ln\frac{x_{i_p}}{x_{j_q}}\right) - \frac{s}{2r(r+s)} \sum_{p=1}^{r} \sum_{q=1}^{r} var\left(ln\frac{x_{i_p}}{x_{i_q}}\right) - \frac{r}{2s(r+s)} \sum_{p=1}^{s} \sum_{q=1}^{s} var\left(ln\frac{x_{j_p}}{x_{j_q}}\right) -$$

$$\frac{r}{2s(r+s)} \sum_{p=1}^{s} \sum_{q=1}^{s} var\left(ln\frac{x_{j_p}}{x_{j_q}}\right) \tag{10}$$

$$COV(z_1, z_2) = \frac{C}{2r_1 s_2} \sum_{p=1}^{r_1} \sum_{q=1}^{s_2} var\left(ln\frac{x_{i_p^1}}{x_{j_q^2}}\right) + \frac{C}{2r_2 s_1} \sum_{p=1}^{r_2} \sum_{q=1}^{s_1} var\left(ln\frac{x_{i_p^2}}{x_{j_q^1}}\right) - \frac{C}{2r_1 r_2} \sum_{p=1}^{r_1} \sum_{q=1}^{r_2} var\left(ln\frac{x_{i_p^1}}{x_{i_q^2}}\right) -$$

$$\frac{C}{2s_1 s_2} \sum_{p=1}^{s_1} \sum_{q=1}^{s_2} var\left(ln\frac{x_{j_p^1}}{x_{j_q^2}}\right), \tag{11}$$

$$CC = \frac{COV(z_1, z_2)}{\sqrt{var(z_1) \cdot var(z_2)}} \tag{12}$$

[revised manuscript text omitted]

balances, we graded each box from 1 to 3,  based on the predicted results and Diebold–Mariano test results—the final results are shown in Fig. 5.  SBP1 performed best in terms of bias, with the lowest ME score of all models, except for GLMRK (Fig. 5a). With respect to the model accuracy assessment, there was no apparent pattern, but the accuracy could be considered hierarchically: (1) for the GLM, SBP3 performed better than the other two SBP methods, and SBP1 performed well when  combined with RK (GLMRK); (2) for RF, SBP1

produced the best result. However, the introduction of RK resulted in the SBP2 and SBP3 performing well (Fig. 5b). In addition, SBP3 and SBP1

delivered a better performance for GLM and RF, respectively (Figs. 5c–5e). Finally, in a comprehensive assessment, SBP1 performed best out of the three SBPs according to SUM6 (Fig. 5f). More details and calculations can be found in the Supplementary Material (Table S4.1).

[Figure]

[Figure]

**Figure 5.** Ranking score of model performance based on three SBPs. SUM1 (a) and SUM2 (b) are the sum scores of ME and RMSE for each model, respectively. SUM3 (c) is the sum score of ME, RMSE and AD for each model; SUM4 (d) and SUM5 (e) are the sum scores of ME or RMSE for GLM$_{all}$ (GLM and GLMRK) and RF$_{all}$ (RF and RFRK); SUM6 (f) is the sum score of all indicators. The lower the value of these scores, the better the model performance.

**3.3 Spatial prediction maps of soil PSFs generated from the different models**

Prediction maps of soil PSFs constructed from the different models are shown in Figs. 6, S3.1, and S3.2. For the components of soil PSFs, the prediction maps of the three components followed a similar rule. The GLM and GLMRK produced broader ranges of predicted values, and their maps were more relevant to the real environment. However, the RF and RFRK predicted a relatively narrow range of low values for these components (sand, silt and clay), revealing a smoother distribution than that generated by the GLM and GLMRK. Unlike the regression methods, the hybrid methods (GLMRK and RFRK ) produced hot and cold spots on the prediction maps and more details of the soil sampling points were apparent (Fig. S5.1).

[Figure]

**(a) GLM_SBP1**    **(b) GLM_SBP2**    **(c) GLM_SBP3**

**(d) GLMRK_SBP1**    **(e) GLMRK_SBP2**    **(f) GLMRK_SBP3**

**(g) RF_SBP1**    **(h) RF_SBP2**    **(i) RF_SBP3**

**(j) RFRK_SBP1**    **(k) RFRK_SBP2**    **(l) RFRK_SBP3**

Sand (%)

- 11    16 - 20    24 - 29    36 - 50
- 16    20 - 24    29 - 36    50 - 75

40    80    160    240    320    km

**Figure 6.** Spatial prediction maps of the sand component of the upper reaches of the Heihe River Basin.

**3.4 Spatial distribution of soil texture classes in the USDA triangles**

The predicted soil textures in the USDA texture triangles (Fig. 7) showed that most predictions fell within the range of observed soil textures (Fig. 3a), and silt loam was the dominant soil texture in all cases. The GLM produced a more discrete distribution than the RF, and the RK method expanded the dispersion. With respect to trends in the predicted samples, the silt components predicted from all models were overestimated. The pattern fitting curves indicated that the prediction results were closer to the bottom right of the USDA triangle than the soil PSF observations. The GLMRK and RFRK curves were longer than the GLM and RF curves, with a more extensive range of values in triangles. Compared with the GLMRK, the RFRK

produced a more upward extension (Figs. 7j,  7l). It was clear that the clay fraction was overestimated and the sand fraction was underestimated.

[Figure]

**Figure 7.** Predicted 262 soil samples in USDA texture triangles using (a) GLM_SBP1, (b) GLM_SBP2, (c) GLM_SBP3, (d) GLMRK_SBP1, (e) GLMRK_SBP2, (f) GLMRK_SBP3, (g) RF_SBP1, (h) RF_SBP2, (i) RF_SBP3, (j) RFRK_SBP1, (k)

RFRK_SBP2, and (l) RFRK_SBP3. Red fitting curves show the trends.

**4 Discussion**

**4.1 Comparison of the GLM, RF, and RK patterns using ILR data**

We found that RF provided more accurate results, but with more bias than the GLM, and that the RK method improved the performance in terms of bias for most models and accuracy of the RF. Odeh et al. (1995) indicated that RK was superior to  linear models, such as MLR, which was reflected in the prediction results for sand in our study. Scarpone et al. (2016)

reported that as a hybrid interpolator, the RFRK outperformed the RF when making soil thickness predictions. We proved that

RFRK was also suitable for compositional data and improved model performance when combined with the ILR

transformation. In summary, the GLM and RF had both advantages and disadvantages when considering the trade-off between bias and accuracy.

The results of GLM and GLMRK should not depend on the choice of ILR basis , which has been proved by previous studies on the use of linear models and kriging for compositional data (Pawlowsky-Glahn et al, 2015). However, the

GLM model used the "glmStepAIC" algorithm (i.e., a stepwise regression) to select the best combination of environmental covariables for each ILR component (Table S2.1). Therefore, the variable inputs are different for these ILR data, and further impact the accuracy assessment and prediction maps. In addition, the difficulty with  GLM is the need for a back- transformation. There is a need to present results on the original untransformed scale after conducting the analysis on a transformed level, which may produce spurious results (Lane, 2002). In our study, we compared the means of ILR transformed data and the original data. We proved the feasibility of the ILR transformation method, especially for meeting the requirements of compositional data. However, the accuracy of the GLM still needs to be improved, which may be because the transformed data did not follow a normal distribution (Fig. 2). With respect to uncertainty, the uncertainty of bias for GLM is higher than that of RF, but the uncertainty of accuracy for GLM is lower. However, RF performed better in terms of accuracy assessment.

Therefore, the main concern was whether the introductions of RK could reduce the uncertainty of RF. With regard to the performances of RFRK and RF, adding RK was recommended in soil PSF interpolation combined with ILR transformed data.

In addition, the range of 95% prediction interval for different models (Figs. S8.1–8.6) demonstrated that the differences were very close. This may because the values of variance for ILR data were small, showing low uncertainty when using ILR

transformed data.

Although the RF had the advantage of prediction accuracy, the limited interpretability of the consequences made it difficult to modify the prediction bias—each tree from the model could not be examined individually (Grimm et al., 2008).

Moreover, the ILR transformation before modeling increased the difficulty of interpretation for not only the predicted values on the ILR scale but also the residuals. The back-transformation of the optimal estimate of log-ratio variables did not generate an optimal estimation of compositional data (Lark and Bishop, 2007), which should also be considered.

**4.2 Comparison of three SBPs of ILR transformation**

Regarding the three SBPs, the ME and RMSE performed better when using SBP1 for ILR transformed data, which may be interpreted as the distributions of the ILR1 and ILR2 of SBP1 being more symmetric (Fig. 2b). In contrast, the performance of SBP2 was worse than that of SBP1 and SBP3 because the ILR_1 component, including all the soil PSF information, was left-skewed (Fig. 2c). This result was especially apparent for the GLM and GLMRK, because the data in a linear model needs to be normally distributed (Lane, 2002).

The negligible difference among the three SBP balances revealed a triangular shape with a cluster at about 120° (Fig. 3b). This could be interpreted as the three soil PSFs having a mixed pattern, with each component dominated by the components in one cluster (Tolosana-Delgado et al., 2005). Although the silt component dominated the soil PSFs (Fig. 2a), sand and clay also played important roles in soil composition. Taking either the most abundant component of the compositional data as the denominator (Martins et al., 2016) or the first component of the permutations did not provide convincing evidence  that the model performs best. This is because using the most abundant component of the compositional data as the primary component of the alterations, i.e., SBP2, resulted in a relatively poor performance compared to the other SBPs. Thus, we recommend that the focus should be on data distribution. Furthermore, the choice of balance and combination of RK are also the key to improving model accuracy, as shown by the result of the RFRK-SBP3 model (Fig. 4).

**4.3 Limitations**

Firstly, the scope of this study is limited to independent modeling. Each ILR component was modeled separately, which may be suboptimal because the components cannot take into account the cross correlations among the ILR coordinates. However, the study has demonstrated the relation of the raw data (sand, silt, and clay) based on ILR transformation, and has confirmed that  currently used prediction models in this work are suitable. In our pervious study, we have used compositional kriging (CK) for the spatial prediction of soil PSFs (Wang and Shi, 2017), and the cross correlations of ILRs can be taken into account using CK. Although CK is optimal, it cannot take into account different balances of ILR, nor can it be combined with a hybrid interpolator (e.g., RK). Moreover, predicting each ILR component separately was a more suitable approach for the spatial prediction models currently used (such as the GLM and RF). Therefore, more alternative spatial prediction models combined with interpretation of ILR balances for compositional data should be considered in the future. For example, CK and high--accuracy surface modelling (HASM; Yue et al., 2007; Yue, 2011; Yue et al., 2016) can be applied for small scale study areas. For large scale study areas, multivariate RF (Segal and Xiao, 2011) can be combined with a log-ratio transformation and hybrid interpolation methods, enabling the cross correlations among ILR coordinates to be better interpreted.

Secondly, the weighting problem was not considered in this study because the ILR method can be qualified as an unweighted log-ratio transformation, giving all parts the same weight for both the definition of the total variance and the reduction of dimensions. This may enlarge the ratios generated from the rare component, which would dominate the analysis (Greenacre and Lewi, 2009). The pairwise log-ratio can be used to set weights by their proportions when there is no knowledge of the component measurement errors (Greenacre, 2019). Nevertheless, all three parts of the soil PSF data dominated the biplot diagram, without the influence of rare elements and with no redundancy; thus, none of the shortcomings mentioned above were apparent. Accuracy assessments using a pairwise log-ratio transformation require further study in the future.

**5 Conclusions**

We evaluated and compared the performances of the GLM, and the RF, and their hybrid patterns (i.e.,

GLMRK and RFRK) using different balances of ILR transformed data. The bias of the GLM was lower than that of the RF; however, the accuracy of the GLM was relatively low. More discrete distributions and broader ranges of prediction value distributions were produced from GLMs in the USDA soil texture triangles. In other words, different predicted data sets were generated from the use of the GLM and RF, with unbiased and inaccurate predictions for the GLM and biased and more accurate predictions for the RF.

The hybrid patterns, GLMRK and RFRK, were found to provide the best solutions because they produced a relatively high prediction accuracy and strong correlations with ECs, providing more details about the soil sampling points (hot spots and cold spots) compared with using the regression model only. However, the non-normal distribution of ILR

data and their residuals, and increased data transformation and inverse transformation , make models more difficult to interpret and improve.

The three SBP-based datasets generated different distributions; a statistical significance test proved that most models had significant differences in prediction accuracy using different SBPs. A ranking score was provided to demonstrate these differences, and compositional balance should be considered when mapping soil PSFs. However, no pattern was apparent, which could be explained by the angle of the biplot diagramwith three rays of soil PSF

components clustered into three modes, and each part dominating its cluster. Using the most abundant component of the compositional data as the first component of the permutations was not considered the best choice for mapping soil PSFs because SBP2 delivered the worst performance. Thus, we recommend that the focus should be on data distribution.

This study provides a reference for the spatial simulation of soil PSFs combined with ECs at the regional scale, and for choosing the balances of ILR transformed data.

*Data Availability.* We did not use any new data and the data we used come from previously published sources. Soil particle- size fractions data is available through our previous studies (Wang and Shi, 2017, 2018). Moreover, it also can be visited on this website: http://data.tpdc.ac.cn/zh-hans/data/7f91d36d-8bbd-40d5-8eaf-7c035e742f40/ (Digital soil mapping dataset of soil texture (soil particle-size fractions) in the upstream of the Heihe river basin (2012-2016); last access: 4 July 2020). The meteorological data can be accessed through http://data.cma.cn/ (last access: 4 July 2020). Environmental covariates data of soil physical and chemical properties and categorical maps can be obtained through http://data.tpdc.ac.cn/zh-hans/ (last access:

4 July 2020), including saturated water content, field water holding capacity, wilt water content, saturated hydraulic conductivity data (http://data.tpdc.ac.cn/zh-hans/data/e977f5e8-972b-42a5-bffe-cd0195f3b42b/, Digital soil mapping dataset of hydrological parameters in the Heihe River Basin (2012); last access: 4 July 2020), and soil thickness data (http://data.tpdc.ac.cn/zh-hans/data/fc84083e-8c66-4a42-b729-4f19334d0d67/, Digital soil mapping dataset of soil depth in the Heihe River Basin (2012-2014); last access: 4 July 2020). DEM data set is provided by the Geospatial Data Cloud site,

Computer Network Information Center, Chinese Academy of Sciences. (http://www.gscloud.cn, last access: 4 July 2020).

*Author contribution.* Wenjiao Shi contributed to soil data sampling, oversaw the design of the entire project. Mo Zhang performed the model analysis and wrote the manuscript. Both authors contributed to writing this paper and interpreting data.

*Competing interests.* The authors declare that they have no conflict of interest.

*Acknowledgment.* Our team expresses gratitude to the following institutions, Key Laboratory of Land Surface Pattern and

Simulation, Institute of Geographic Sciences and Natural Resources Research, Chinese Academy of Sciences; School of Earth

Sciences and Resources, China University of Geosciences; College of Resources and Environment, University of Chinese

Academy of Sciences. This study was supported by the National Natural Science Foundation of China (Grant No. 41771111

and 41930647), the Youth Innovation Promotion Association, CAS (No. 2018071), and a grant from State Key Laboratory of

Resources and Environmental Information System. This study was supported by the National Key Research and Development

[revised manuscript text omitted]

---

## Author Comment (AC2)

**Responses to the Referee**

**General comments:**

**The topic of the paper is relevant, but the study has limitations and weaknesses that should be tackled.**

**Specific comments:**

**Comment 1: The literature review in the Introduction is wide, but not well organized and structured. I suggest reorganizing and rephrasing it to improve readability.**

**Response:** Thanks for the referee's suggestion for the Introduction. We have reorganized this section in our revised version.

**Comment 2: In the Introduction it is mentioned that "It has been proven in statistical science that different results are obtained using different choices of ILR balances", but no references are provided to support this claim. Please, add references of works where the influence of the choice of ILR balances on the results is demonstrated.**

**Response:** Thanks for the referee's suggestion. We have added the references and introduced more details about the choice of ILR balances.

**P3L85:** *"It has been proven in statistical science that different results are obtained using different ILR balances. For example, Fiserova and Hron (2011) reported that different balances generated different covariance structures. Moreover, the choice of SBP is related to hypotheses, research questions of interest, or the context of the data analysis (Coenders et al., 2017; Facevicova et al., 2018). Thus, the option of a specific SBP for compositions is crucial for the intended interpretation of*
*coordinates (Fiserova and Hron, 2011)."*

**Comment 3: In equations (7) and (8), are M_i and P_i (D-1)- vectors? If so, the expressions (7) and (8) should account for that. Otherwise, their meaning should be better explained.**

**Response:** Thanks for the referee's suggestion for the equations. $M_i$ and $P_i$ are not D-1 vectors, instead, they are measured
and predicted values for raw data (i.e., sand, silt and clay), respectively. This is because we used these indicators (ME, RMSE) to measure prediction bias and accuracy compared with the raw data (D), $P_i$ is the predicted value that back-transformed to the real space. Therefore, $M_i$ and $P_i$ are D vectors (D = 3) for sand, silt and clay. We have improved this part in our revised version.

**P8L218:** *"where $M_i$ and $P_i$ are the measured and predicted values of the ith sample for sand, silt and clay."*

**Comment 4: Are the differences found in the indicators significant? The comparisons reported (for example in Sections 3.2, 4.1 and 4.2) should be based on statistical tests to verify the statistical significance of the differences observed in the indicators. The significance of the differences should also be taken into consideration in the grading of the three**

I

**SBP at the end of Section 3.2, otherwise the results shown in Figure 5 may be misleading.**

**Response:** Thank you for your suggestion regarding the statistical significance test for the results of different models. Based on a 30-times independent dataset validation method, we performed a Diebold–Mariano test (Diebold and Mariano, 1995; Harvey et al., 1997) for the two models. The $p$-values of MEs and RMSEs of sand, silt, and clay fractions, and also Ads, were shown in the following tables. The null hypothesis is that the two methods have the same level of prediction accuracy, and the alternative hypothesis is that method 1 and method 2 have different levels of accuracy (when $p<0.05$, the null hypothesis is rejected). The significance tests showed that most $p$-values were significant, expect for the ME of sand fractions. The differences were reflected in not only different models (GLM and RF), but also in different SBPs when using the same model. For the RF model, combining RK or not is not significant, which may be because RF has less uncertainty, or because the RF algorithm is robust to ILR data and residuals. For Fig. 5, we used the mean value of the 30-times results. We have adopted your suggestion and recalculated these indicators based on the significance test. The basic principle is that if the results of different SBPs were not significant, then the same score was used; if they were significant, different scores were used based on model performance. The new result was shown in Fig. 5, and we found that different SBPs delivered different performances in GLM and RF. In bias assessment, SBP1 performed best. Regarding accuracy, SBP3 and SBP1 delivered a good performance for GLM and RF, respectively. Moreover, in the comprehensive assessment, SBP1 still performed best out of the three SBPs according to SUM6, which is the same as we did before. In summary, this evaluation method is designed to evaluate the performance of different SBPs based on the predicted results and significance test. Using the ranking score in Fig. 5, we demonstrated that different SBP balances generated significant differences, and compositional balance should be considered when mapping soil PSFs. We also included this in the Conclusion Section. Therefore, we have improved this part in our revised version and Supplementary Material.

**P8L204:** *"Moreover, the Diebold–Mariano test (Diebold and Mariano, 1995; Harvey et al., 1997) was used to verify the statistical significance of the differences among the models."*

**P11L276:** *"To assess the accuracy of the different models, the Diebold–Mariano test was used, which showed that the statistical differences of most models were significant. This significance was reflected not only in different models (GLM and RF), but also in different SBPs when using the same model (Tables S6.1–S6.7)."*

**P12L294:** *"The model performances were different for the three SBPs. To better evaluate model performance using the different SBP balances, we graded each box from 1 to 3 based on the predicted results and Diebold¬–Mariano test results— the final results are shown in Fig. 5. SBP1 performed best in terms of bias, with the lowest ME score of all models, except for GLMRK (Fig. 5a). With respect to the model accuracy assessment, there was no apparent pattern, but the accuracy could be considered hierarchically: (1) for the GLM, SBP3 performed better than the other two SBP methods, and SBP1 performed well when combined with RK (GLMRK); (2) for RF, SBP1 produced the best result. However, the introduction of RK resulted*

*in the SBP2 and SBP3 performing well (Fig. 5b). In addition, SBP3 and SBP1 delivered a better performance for GLM and RF, respectively (Figs. 5c–5e). Finally, in a comprehensive assessment, SBP1 performed best out of the three SBPs according to SUM6 (Fig. 5f). More details and calculations can be found in the Supplementary Material (Table S4.1)."*

[Figure]

*"**Figure 5.** Ranking score of model performance based on three SBPs. SUM1 (a) and SUM2 (b) are the sum scores of ME and RMSE for each model, respectively. SUM3 (c) is the sum score of ME, RMSE, and AD for each model; SUM4 (d) and SUM5 (e) are the sum scores of ME or RMSE for GLM$_{all}$ (GLM and GLMRK) and RF$_{all}$ (RF and RFRK); SUM6 (f) is the sum score of all indicators. The lower the value of these scores, the better the model performance."*

**P20L415:** *"The three SBP-based datasets generated different distributions; a statistical significance test proved that most models had significant differences in prediction accuracy using different SBPs. A ranking score was provided to demonstrate these differences, and compositional balance should be considered when mapping soil PSFs."*

**Reference**

Diebold, F.X. and Mariano, R.S.: Comparing predictive accuracy. Journal of Business and Economic Statistics, 13, 253-263,
1995.

Harvey, D., Leybourne, S., and Newbold, P.: Testing the equality of prediction mean squared errors. International Journal of forecasting, 13(2), 281-291, 1997.

**Table S6.1.** The statistical significance *p*-value (Diebold-Mariano test) of predicted ME of sand fractions among different models.

| ME_Sand | GLM_SB P1 | GLM_SB P2 | GLM_SB P3 | GLMRK_SB P1 | GLMRK_SB P2 | GLMRK_SB P3 | RF_SBP 1 | RF_SBP 2 | RF_SBP 3 | RFRK_SB P1 | RFRK_SB P2 | RFRK_SB P3 |
|---|---|---|---|---|---|---|---|---|---|---|---|---|
| GLM_SBP1 | - | | | | | | | | | | | |
| GLM_SBP2 | 0.07 | - | | | | | | | | | | |
| GLM_SBP3 | 0.85 | 0.22 | - | | | | | | | | | |
| GLMRK_SB P1 | 0.83 | 0.19 | 0.98 | - | | | | | | | | |
| GLMRK_SB P2 | 0.18 | 0.67 | 0.20 | 0.02** | - | | | | | | | |
| GLMRK_SB P3 | 0.84 | 0.31 | 0.90 | 0.89 | 0.03** | - | | | | | | |
| RF_SBP1 | 0.06 | 0.61 | 0.04** | 0.08 | 0.89 | 0.15 | - | | | | | |
| RF_SBP2 | 0.02** | 0.39 | 0.02** | 0.04** | 0.71 | 0.12 | 0.25 | - | | | | |
| RF_SBP3 | 0.02** | 0.35 | 0.02** | 0.03** | 0.61 | 0.09 | 0.08 | 0.41 | - | | | |
| RFRK_SBP1 | 0.39 | 0.90 | 0.38 | 0.39 | 0.93 | 0.46 | 0.83 | 0.70 | 0.63 | - | | |
| RFRK_SBP2 | 0.28 | 0.71 | 0.28 | 0.30 | 0.89 | 0.37 | 0.95 | 0.92 | 0.84 | 0.20 | - | |
| RFRK_SBP3 | 0.33 | 0.76 | 0.32 | 0.32 | 0.93 | 0.39 | 0.99 | 0.88 | 0.81 | 0.29 | 0.72 | - |

*Note that $p>0.05$: not significant; $0.01<p<0.05$: significant; $p<0.01$: highly significant. The null hypothesis is that the two methods have the same level of prediction accuracy, and the alternative hypothesis is that method 1 and method 2 have different levels of accuracy (when $p<0.05$, null hypothesis rejected).

**Table S6.2.** The statistical significance *p*-value (Diebold-Mariano test) of predicted ME of silt fractions among different models.

| ME_Silt | GLM_SB P1 | GLM_SB P2 | GLM_SB P3 | GLMRK_SB P1 | GLMRK_SB P2 | GLMRK_SB P3 | RF_SBP 1 | RF_SBP 2 | RF_SBP 3 | RFRK_SB P1 | RFRK_SB P2 | RFRK_SB P3 |
|---|---|---|---|---|---|---|---|---|---|---|---|---|
| GLM_SBP1 | - | | | | | | | | | | | |
| GLM_SBP2 | 0.32 | - | | | | | | | | | | |
| GLM_SBP3 | 0.72 | 0.63 | - | | | | | | | | | |
| GLMRK_SBP1 | 0.03** | 0*** | 0.01** | - | | | | | | | | |
| GLMRK_SBP2 | 0.40 | 0.07 | 0.23 | 0.16 | - | | | | | | | |
| GLMRK_SBP3 | 0.19 | 0.04** | 0.05 | 0.83 | 0.19 | - | | | | | | |
| RF_SBP1 | 0*** | 0*** | 0*** | 0*** | 0*** | 0*** | - | | | | | |
| RF_SBP2 | 0*** | 0*** | 0*** | 0*** | 0*** | 0*** | 0.01** | - | | | | |
| RF_SBP3 | 0*** | 0*** | 0*** | 0*** | 0*** | 0*** | 0*** | 0.67 | - | | | |
| RFRK_SBP1 | 0.01** | 0.02*** | 0.01** | 0*** | 0*** | 0*** | 0.44 | 0.23 | 0.20 | - | | |
| RFRK_SBP2 | 0.01** | 0.01** | 0.01** | 0*** | 0*** | 0*** | 0.72 | 0.44 | 0.39 | 0.12 | - | |
| RFRK_SBP3 | 0.01** | 0.01** | 0*** | 0*** | 0*** | 0*** | 0.85 | 0.54 | 0.49 | 0.03** | 0.36 | - |

*Note that $p>0.05$: not significant; $0.01<p<0.05$: significant; $p<0.01$: highly significant. The null hypothesis is that the two methods have the same level of prediction accuracy, and the alternative hypothesis is that method 1 and method 2 have different levels of accuracy (when $p<0.05$, null hypothesis rejected).

**Table S6.3.** The statistical significance $p$-value (Diebold-Mariano test) of predicted ME of clay fractions among different models.

| ME_Clay | GLM_SB P1 | GLM_SB P2 | GLM_SB P3 | GLMRK_SB P1 | GLMRK_SB P2 | GLMRK_SB P3 | RF_SBP 1 | RF_SBP 2 | RF_SBP 3 | RFRK_SB P1 | RFRK_SB P2 | RFRK_SB P3 |
|---|---|---|---|---|---|---|---|---|---|---|---|---|
| GLM_SBP1 | - | | | | | | | | | | | |
| GLM_SBP2 | 0.06 | - | | | | | | | | | | |
| GLM_SBP3 | 0.07 | 0.95 | - | | | | | | | | | |
| GLMRK_SB P1 | 0*** | 0*** | 0*** | - | | | | | | | | |
| GLMRK_SB P2 | 0.02** | 0*** | 0*** | 0.39 | - | | | | | | | |
| GLMRK_SB P3 | 0.04** | 0*** | 0*** | 0.12 | 0.55 | - | | | | | | |
| RF_SBP1 | 0*** | 0*** | 0*** | 0*** | 0*** | 0*** | - | | | | | |
| RF_SBP2 | 0*** | 0*** | 0*** | 0*** | 0*** | 0*** | 0.04** | - | | | | |
| RF_SBP3 | 0*** | 0*** | 0*** | 0*** | 0*** | 0*** | 0.04** | 0.64 | - | | | |
| RFRK_SBP1 | 0.03** | 0.15 | 0.11 | 0*** | 0*** | 0*** | 0.36 | 0.26 | 0.28 | - | | |
| RFRK_SBP2 | 0.03** | 0.14 | 0.11 | 0*** | 0*** | 0*** | 0.35 | 0.26 | 0.27 | 0.97 | - | |
| RFRK_SBP3 | 0.02** | 0.09 | 0.07 | 0*** | 0*** | 0*** | 0.56 | 0.43 | 0.45 | 0.01** | 0.01** | - |

*Note that $p>0.05$: not significant; $0.01<p<0.05$: significant; $p<0.01$: highly significant. The null hypothesis is that the two methods have the same level of prediction accuracy, and the alternative hypothesis is that method 1 and method 2 have different levels of accuracy (when p<0.05, null hypothesis rejected).

**Table S6.4.** The statistical significance *p*-value (Diebold-Mariano test) of predicted RMSE of sand fractions among different models.

| RMSE_Sand | GLM_SBP 1 | GLM_SBP 2 | GLM_SBP 3 | GLMRK_SBP 1 | GLMRK_SBP 2 | GLMRK_SBP 3 | RF_SBP 1 | RF_SBP 2 | RF_SBP 3 | RFRK_SBP 1 | RFRK_SBP 2 | RFRK_SBP 3 |
|---|---|---|---|---|---|---|---|---|---|---|---|---|
| GLM_SBP1 | - | | | | | | | | | | | |
| GLM_SBP2 | 0*** | - | | | | | | | | | | |
| GLM_SBP3 | 0.42 | 0.02** | - | | | | | | | | | |
| GLMRK_SBP 1 | 0.02** | 0*** | 0.01** | - | | | | | | | | |
| GLMRK_SBP 2 | 0.03** | 0.07 | 0.18 | 0*** | - | | | | | | | |
| GLMRK_SBP 3 | 0.94 | 0.02** | 0.53 | 0.01 | 0.08 | - | | | | | | |
| RF_SBP1 | 0.01** | 0*** | 0.01** | 0.77 | 0*** | 0.1 | - | | | | | |
| RF_SBP2 | 0.02** | 0*** | 0.01** | 0.76 | 0*** | 0.09 | 0.92 | - | | | | |
| RF_SBP3 | 0.07 | 0*** | 0.04** | 0.88 | 0.01** | 0.22 | 0*** | 0.01** | - | | | |
| RFRK_SBP1 | 0.16 | 0*** | 0.12 | 0.61 | 0.02** | 0.19 | 0.72 | 0.72 | 0.54 | - | | |
| RFRK_SBP2 | 0.26 | 0.01** | 0.19 | 0.79 | 0.03** | 0.28 | 0.92 | 0.92 | 0.72 | 0.05** | - | |
| RFRK_SBP3 | 0.23 | 0.01** | 0.16 | 0.74 | 0.03** | 0.25 | 0.87 | 0.88 | 0.67 | 0.08 | 0.61 | - |

*Note that $p>0.05$: not significant; $0.01<p<0.05$: significant; $p<0.01$: highly significant. The null hypothesis is that the two methods have the same level of prediction accuracy, and the alternative hypothesis is that method 1 and method 2 have different levels of accuracy (when $p<0.05$, null hypothesis rejected).

**Table S6.5.** The statistical significance *p*-value (Diebold-Mariano test) of predicted RMSE of silt fractions among different models.

| RMSE_Silt | GLM_SB P1 | GLM_SB P2 | GLM_SB P3 | GLMRK_SB P1 | GLMRK_SB P2 | GLMRK_SB P3 | RF_SBP 1 | RF_SBP 2 | RF_SBP 3 | RFRK_SB P1 | RFRK_SB P2 | RFRK_SB P3 |
|---|---|---|---|---|---|---|---|---|---|---|---|---|
| GLM_SBP1 | - | | | | | | | | | | | |
| GLM_SBP2 | 0.44 | - | | | | | | | | | | |
| GLM_SBP3 | 0.03** | 0.01** | - | | | | | | | | | |
| GLMRK_SB P1 | 0.01** | 0.01** | 0.98 | - | | | | | | | | |
| GLMRK_SB P2 | 0.50 | 0.80 | 0.02** | 0*** | - | | | | | | | |
| GLMRK_SB P3 | 0.24 | 0.12 | 0.70 | 0.67 | 0.02 | - | | | | | | |
| RF_SBP1 | 0*** | 0*** | 0*** | 0*** | 0*** | 0*** | - | | | | | |
| RF_SBP2 | 0*** | 0*** | 0*** | 0*** | 0*** | 0*** | 0*** | - | | | | |
| RF_SBP3 | 0*** | 0*** | 0*** | 0*** | 0*** | 0.01** | 0*** | 0.01 | - | | | |
| RFRK_SBP1 | 0*** | 0*** | 0*** | 0*** | 0*** | 0*** | 0.27 | 0.10 | 0.05** | - | | |
| RFRK_SBP2 | 0*** | 0*** | 0*** | 0*** | 0*** | 0*** | 0.69 | 0.36 | 0.23 | 0*** | - | |
| RFRK_SBP3 | 0*** | 0*** | 0*** | 0*** | 0*** | 0*** | 0.54 | 0.24 | 0.14 | 0.01** | 0.36 | - |

*Note that $p>0.05$: not significant; $0.01<p<0.05$: significant; $p<0.01$: highly significant. The null hypothesis is that the two methods have the same level of prediction accuracy, and the alternative hypothesis is that method 1 and method 2 have different levels of accuracy (when p<0.05, null hypothesis rejected).

**Table S6.6.** The statistical significance *p*-value (Diebold-Mariano test) of predicted RMSE of clay fractions among different models.

| RMSE_Clay | GLM_SB P1 | GLM_SB P2 | GLM_SB P3 | GLMRK_SB P1 | GLMRK_SB P2 | GLMRK_SB P3 | RF_SBP 1 | RF_SBP 2 | RF_SBP 3 | RFRK_SB P1 | RFRK_SB P2 | RFRK_SB P3 |
|---|---|---|---|---|---|---|---|---|---|---|---|---|
| GLM_SBP1 | - | | | | | | | | | | | |
| GLM_SBP2 | 0.05** | - | | | | | | | | | | |
| GLM_SBP3 | 0.02** | 0.45 | - | | | | | | | | | |
| GLMRK_SB P1 | 0.32 | 0.05** | 0.01** | - | | | | | | | | |
| GLMRK_SB P2 | 0.50 | 0.65 | 0.34 | 0.01** | - | | | | | | | |
| GLMRK_SB P3 | 0.84 | 0.39 | 0.16 | 0.66 | 0.42 | - | | | | | | |
| RF_SBP1 | 0*** | 0*** | 0*** | 0*** | 0*** | 0*** | - | | | | | |
| RF_SBP2 | 0*** | 0*** | 0*** | 0*** | 0*** | 0*** | 0*** | - | | | | |
| RF_SBP3 | 0*** | 0*** | 0*** | 0*** | 0*** | 0*** | 0.84 | 0*** | - | | | |
| RFRK_SBP1 | 0*** | 0*** | 0*** | 0*** | 0*** | 0*** | 0.74 | 0.29 | 0.72 | - | | |
| RFRK_SBP2 | 0*** | 0*** | 0*** | 0*** | 0*** | 0*** | 0.67 | 0.27 | 0.66 | 0.59 | - | |
| RFRK_SBP3 | 0*** | 0*** | 0*** | 0*** | 0*** | 0*** | 0.63 | 0.23 | 0.60 | 0.50 | 0.87 | - |

*Note that $p>0.05$: not significant; $0.01<p<0.05$: significant; $p<0.01$: highly significant. The null hypothesis is that the two methods have the same level of prediction accuracy, and the alternative hypothesis is that method 1 and method 2 have different levels of accuracy (when p<0.05, null hypothesis rejected).

**Table S6.7.** The statistical significance *p*-value (Diebold-Mariano test) of predicted AD among different models.

| AD | GLM_SB P1 | GLM_SB P2 | GLM_SB P3 | GLMRK_SB P1 | GLMRK_SB P2 | GLMRK_SB P3 | RF_SBP 1 | RF_SBP 2 | RF_SBP 3 | RFRK_SB P1 | RFRK_SB P2 | RFRK_SB P3 |
|---|---|---|---|---|---|---|---|---|---|---|---|---|
| GLM_SBP1 | - | | | | | | | | | | | |
| GLM_SBP2 | 0.92 | - | | | | | | | | | | |
| GLM_SBP3 | 0.08 | 0.08 | - | | | | | | | | | |
| GLMRK_SB P1 | 0.63 | 0.70 | 0.41 | - | | | | | | | | |
| GLMRK_SB P2 | 0.97 | 0.98 | 0.25 | 0.31 | - | | | | | | | |
| GLMRK_SB P3 | 0.77 | 0.81 | 0.42 | 0.95 | 0.70 | - | | | | | | |
| RF_SBP1 | 0*** | 0*** | 0*** | 0*** | 0*** | 0*** | - | | | | | |
| RF_SBP2 | 0*** | 0*** | 0*** | 0*** | 0*** | 0*** | 0*** | - | | | | |
| RF_SBP3 | 0*** | 0*** | 0*** | 0*** | 0*** | 0*** | 0.06 | 0.2 | - | | | |
| RFRK_SBP1 | 0*** | 0*** | 0*** | 0*** | 0*** | 0*** | 0.28 | 0.12 | 0.17 | - | | |
| RFRK_SBP2 | 0*** | 0*** | 0*** | 0*** | 0*** | 0*** | 0.38 | 0.17 | 0.23 | 0.34 | - | |
| RFRK_SBP3 | 0*** | 0*** | 0*** | 0*** | 0*** | 0*** | 0.38 | 0.18 | 0.23 | 0.32 | 0.95 | - |

*Note that $p>0.05$: not significant; $0.01<p<0.05$: significant; $p<0.01$: highly significant. The null hypothesis is that the two methods have the same level of prediction accuracy, and the alternative hypothesis is that method 1 and method 2 have different levels of accuracy (when $p<0.05$, null hypothesis rejected).

X

**Comment 5: The authors should carefully check the English for grammar errors throughout the manuscript. A serious editing and proof-reading are required.**

**Response:** Thanks for the referee's suggestion about the English quality of this paper. We looked for some senior editors from a professional English polishing company to improve the overall language of this article and we have checked and improved the writing in the revised version. The editorial certificate is shown as follows:

[Figure]

**EDITORIAL CERTIFICATE**

This document certifies that the manuscript below was edited for correct
English language usage, grammar, punctuation and spelling by qualified
native English speaking editors at Charlesworth Author Services.

**Paper Title:**

Compositional balance should be considered in soil particle-size fractions
mapping using hybrid interpolators

**Author:**

Wenjiao Shi

**Date certificate issued:**

August 16, 2021

cwauthors.com

**Compositional balance should be considered in the mapping of soil particle-size fractions using hybrid interpolators**

Mo Zhang[1,2], Wenjiao Shi[1,2]

[1]Key Laboratory of Land Surface Pattern and Simulation, State Key Laboratory of Resources and Environmental Information System, Institute of Geographic Sciences and Natural Resources Research, Chinese Academy of Sciences, Beijing 100101, China

[2]College of Resources and Environment, University of Chinese Academy of Sciences, Beijing 100049, China

*Correspondence to:* Wenjiao Shi (shiwj@lreis.ac.cn), Institute of Geographic Sciences and Natural Resources Research, Chinese Academy of Sciences. 11A, Datun Road, Chaoyang District, Beijing 100101, China.

**Abstract**. Digital soil mapping of soil particle-size fractions (PSFs) using log-ratio methods is a widely used technique. As a hybrid interpolator, regression kriging (RK) is one way to improve prediction accuracy of soil PSFs. However, there is still a lack of comparisons and recommendations when RK is applied for compositional data. It is unknown if the prediction performance based on different balances of the isometric log-ratio (ILR) transformation is robust. We compared the generalized linear model (GLM), the random forest (RF) model, and their hybrid patterns (i.e., GLMRK and RFRK) using different transformed data based on three ILR balances. The comparison involved 29 environmental covariables  for the soil PSF prediction in the upper reaches of the Heihe River Basin, China. The results showed that RF performed best, with more accurate predictions, but GLM produced a more unbiased prediction. For the hybrid interpolators, RK was recommended because it widened the data ranges of the prediction values and modified the bias and accuracy for most models, especially for RF. Moreover, prediction maps generated from RK revealed more details of the soil sampling points. For three ILR balances, different data distributions were produced. Using the most abundant component of the compositional data as the first component of the permutations was not considered the best choice for soil PSF mapping. Compared to the relative abundance of components, we recommend that the focus should be on data distribution. This study provides a reference for mapping  soil PSFs combined with transformed data at the regional scale.

**1 Introduction**

Recently, spatial interpolation of soil particle-size fractions (PSFs) has become a focus of soil science researchers. More accurately predicted soil PSFs elucidates hydrological, physical, and environmental processes (Delbari et al., 2011; Ließ et al., 2012; McBratney et al., 2002).

The characteristics of compositional data make soil PSFs more impressive than other soil properties. Soil PSFs are usually expressed as three components for discrete data sand, silt, and clay and carry only relevant percentage information. Soil texture is classified by soil PSFs, which can be demonstrated on a ternary diagram (so-called soil texture triangle). However, the closure system formed in this triangle is not in Euclidean space, but rather Aitchison space (i.e., a simplex) (Aitchison, 1986). Due to "spurious correlations" (Pawlowsky-Glahn, 1984), traditional statistical methods based on Euclidean geometry may generate mistakes when dealing directly with soil PSF data (Filzmoser et al., 2009). The requirement for constant-sum, nonnegative, unbiased prediction is the key to spatial interpolation (Walvoort and de Gruijter, 2001). Therefore, data transformation from a simplex into real space is crucial for compositional data . Log ratio transformations play a significant role in compositional data analysis, including the additive log-ratio (ALR), centered log-ratio (CLR) (Aitchison, 1986), and the isometric log-ratio (ILR) (Egozcue et al., 2003).

Although these three log-ratio methods have been widely applied to transform soil PSF data, different study area scales and model selection should be considered when modeling. For local-scale study areas, geostatistical models combined with log-ratio transformed data are sufficient to map spatial patterns, as shown in our previous study (Wang and Shi, 2017). From another perspective, functional compositions combined with the kriging method can also be applied to produce soil particle size curves (PSCs) (Menafoglio et al., 2014), providing an abundance of information. Functional compositions involve the use of complete and continuous information rather than discrete information, and soil PSFs can be extracted from the predicted soil PSCs (Menafoglio et al., 2016a). Log-ratio transformations can also be combined with functional-compositional data for the stochastic simulation of PSCs (Menafoglio et al., 2016b, Talska et al., 2018). For middle-scale study areas, outliers may lead to overestimation of the variogram, resulting in prediction errors (Lark, 2000). Therefore, spatial interpolation should take robust variogram estimators into account to improve model performance (Lark, 2003). A previous study proved that applying robust variogram estimators in log-ratio co-kriging significantly improved mapping performance (Wang and Shi, 2018). For large-scale study areas, geostatistical models are limited by the number of soil samples and increased spatial variability. Increasing numbers of studies have concentrated on mapping soil PSFs using different machine-learning models combined with ancillary data (i.e., environmental covariables, ECs). Log-ratio transformed data have been applied on a broad basin scale (Zhang et al., 2020), national scale (Akpa et al., 2014), and even a global scale (Hengl et al., 2017).

Among these EC-combined models, linear, machine-learning, geostatistical models, and high-accuracy surface modeling (Yue et al., 2020; Shi et al., 2016) have been commonly used in middle or large-scale studies. Linear models, for example, the generalized linear model (GLM) and multiple linear regression (MLR) model, have been used in soil PSF predictions because of their flexibility and interpretability (Lane, 2002; Buchanan et al., 2012). Many machine-learning models have been applied for soil PSF interpolation and soil texture classification. For example, tree learners, such as the random forest (RF) model, are advantageous due to their ability to handle overfitting and generate more realistic maps (Zhang et al., 2020). In addition, regression kriging (RK) has proven to be a powerful and widely accepted method for soil interpolation. ECs can be introduced through its regression function and improved model accuracy as a hybrid interpolator for some soil properties, such as topsoil thickness and pH (Hengl et al., 2004; Keskin and Grunwald, 2018; Shi et al., 2009; Shi et al., 2011). However, the scope of the comparison needs to be expanded to further explore the prediction accuracy combined with compositional data using linear models, machine-learning models, and combining RK with other models (hybrid patterns).

In log-ratio methods, the ILR method performs better than the ALR and CLR methods in both theory and  practice (Filzmoser and Hron, 2009; Wang and Shi, 2018; Zhang et al., 2020). The ILR method eliminates model collinearity and preserves advantageous properties. such as isometry, scale invariance, and sub-compositional coherence.

The ILR method is constructed in orthonormal coordinate systems (i.e., balances) using a sequential binary partition (SBP)

(Egozcue and Pawlowsky-Glahn, 2005). The choices of balances are not unique. Multiple sets of ILR

transformed data can be generated by permutations of components (different SBPs) in the compositional data. The selection of an SBP can be based on prior expert knowledge, using a compositional biplot (Lloyd et al., 2012) or variograms and cross-variograms (Molayemat et al., 2018). It has been proven in statistical science that different results are obtained using different ILR balances. For example, Fiserova and Hron (2011) reported that different balances generated different covariance structures. Moreover, the choice of SBP is related to hypotheses, research questions of interest, or the context of the data analysis (Coenders et al., 2017; Facevicova et al., 2018). Thus, 
[revised manuscript text omitted]

**2.5 Prediction method system and validation**

The method  of spatial interpolation  for soil PSFs is presented in Table 2. We systematically compared 12 models: the combinations of four interpolators,  (GLM , GLMRK, RF , RFRK), and three SBPs of the ILR transformation method. For the validation of model performance,  independent data set validation was used to evaluate the prediction bias and accuracy of the models. The  data were randomly divided into two sets: the sub-training sets (70%) and the sub-testing sets (30%), and this process was repeated 30 times. Moreover, the Diebold–Mariano test (Diebold and Mariano, 1995; Harvey et al., 1997) was used to verify the statistical significance of the differences among the models.

**Table 2.** The method  of spatial interpolation  of soil PSFs.

| Models | GLM | GLMRK | RF | RFRK |
|---|---|---|---|---|
|  SBP1 | GLM_SBP1 | GLMRK_SBP1 | RF_SBP1 | RFRK_SBP1 |
|  SBP2 | GLM_SBP2 | GLMRK_SBP2 | RF_SBP2 | RFRK_SBP2 |
|  SBP3 | GLM_SBP3 | GLMRK_SBP3 | RF_SBP3 | RFRK_SBP3 |

The mean error (ME), the root mean square error (RMSE), and Aitchison distance (AD) were used to evaluate and compare the prediction performance. The ME and RMSE measure prediction bias and accuracy, respectively (Odeh et al., 1995). The AD is an overall indicator of compositional analysis, which describes the distance between two compositions. Generally, in an accurate, unbiased model all three values will be close to 0. The ranges of a 95% confidence interval (CI) (Streiner, 1996) of ME, RMSE, and AD were calculated to compare the uncertainties of different models. The ME, RMSE, and AD were calculated as follows:

$$ME = \frac{1}{n}\sum_{i=1}^{n}(M_i - P_i), \tag{7}$$

$$RMSE = \sqrt{\frac{1}{n}\sum_{i=1}^{n}(M_i - P_i)^2}, \tag{8}$$

$$AD = \left[\sum_{i=1}^{D}\left(log\frac{M_i}{G(\boldsymbol{M})} - log\frac{P_i}{G(\boldsymbol{P})}\right)^2\right]^{0.5}, \tag{9}$$

where $M_i$ and $P_i$ are the measured and predicted values of the $i$th sample for sand, silt and clay; $n$ refers to the number of soil samples; $D$ is the number of dimensions of compositions; and $G(\boldsymbol{M})$ and $G(\boldsymbol{P})$ denote the geometric mean with the form $G(\mathbf{x}) = (x_1, \ldots, x_D)^{1/D}$ of the measured and predicted values, respectively.

**2.6 Covariance structure analysis**

The interpretation of the ILR balances is based on a decomposition of the covariance (COV) structure (Fiserova and Hron, 2011). We calculated the variance (VAR), COV, and the corresponding correlation coefficient (CC) of ILR transformed data based on different SBPs. The equations for calculating VAR, COV, and CC were derived from Eq. (1) as follows:

$$VAR(z) = \frac{1}{r+s} \sum_{p=1}^{r} \sum_{q=1}^{s} var\left(ln\frac{x_{i_p}}{x_{j_q}}\right) - \frac{s}{2r(r+s)} \sum_{p=1}^{r} \sum_{q=1}^{r} var\left(ln\frac{x_{i_p}}{x_{i_q}}\right) - \frac{r}{2s(r+s)} \sum_{p=1}^{s} \sum_{q=1}^{s} var\left(ln\frac{x_{j_p}}{x_{j_q}}\right) -$$

$$\frac{r}{2s(r+s)} \sum_{p=1}^{s} \sum_{q=1}^{s} var\left(ln\frac{x_{j_p}}{x_{j_q}}\right) \tag{10}$$

$$COV(z_1, z_2) = \frac{C}{2r_1 s_2} \sum_{p=1}^{r_1} \sum_{q=1}^{s_2} var\left(ln\frac{x_{i_p^1}}{x_{j_q^2}}\right) + \frac{C}{2r_2 s_1} \sum_{p=1}^{r_2} \sum_{q=1}^{s_1} var\left(ln\frac{x_{i_p^2}}{x_{j_q^1}}\right) - \frac{C}{2r_1 r_2} \sum_{p=1}^{r_1} \sum_{q=1}^{r_2} var\left(ln\frac{x_{i_p^1}}{x_{i_q^2}}\right) -$$

$$\frac{C}{2s_1 s_2} \sum_{p=1}^{s_1} \sum_{q=1}^{s_2} var\left(ln\frac{x_{j_p^1}}{x_{j_q^2}}\right), \tag{11}$$

$$CC = \frac{COV(z_1, z_2)}{\sqrt{var(z_1) \cdot var(z_2)}} \tag{12}$$

For soil PSF data, Eqs. (10 (12) can be simplified to three dimensions. The relationship between the ratios of soil PSF components and the dominant roles of ILR transformed data were indicated from the covariance structure. All  statistical analyses, such as the descriptive statistics of soil PSF data, calculation and evaluation of indicators, and the spatial prediction mapping, were performed using the R statistical program (R Development Core Team, 2019).

**3 Results**

**3.1 Exploratory data analysis**

**3.1.1 Descriptive statistics of soil PSF data**

From the descriptive statistics of the original (raw) and ILR transformed data, the silt fraction dominated the soil PSFs, larger than the sand and clay fractions. The distributions of the sand and clay fractions were similar (Fig. 2a). The ILR transformed data based on the three SBPs revealed different distributions (Figs. 2b

2d). For example, two ILR components (ILR1 and ILR2) for SBP1 had a symmetric distribution around zero on the *x*-axis (Fig. 2b). In comparison, the distribution of data generated from SBP2 and SBP3 had a mirrored symmetry, with a left- skewed ILR1 of SBP2 and right-skewed ILR2 of SBP3 (Figs. 2c and 2d). A comparison of means and medians demonstrated that the back-transformed means of three sets of ILR transformed data were the same, and the mean ILR of sand was closer to the median of sand compared with the original soil PSF data. In contrast,  opposite patterns were apparent for the silt and clay components (Fig. 2e).

[Figure]

| Center | Sand | Silt | Clay | Sand_ILR | Silt_ILR | Clay_ILR |
|--------|------|------|------|----------|----------|----------|
| Mean | 19.61 | 66.80 | 13.59 | 18.07 | 71.48 | 10.45 |
| Median | 17.03 | 67.44 | 14.53 | — | — | — |

**Figure 2.** Descriptive statistics of original soil PSF and ILR transformed data using different SBPs. Note that the means of

Sand_ILR, Silt_ILR, and Clay_ILR from different SBPs were back-transformed into real space.

**3.1.2 Covariance structure of ILR transformed data with different balances**

Covariance analysis of the transformed data of soil PSFs based on the different SBPs showed that the variance VarILR_1 of SBP3 was the largest, followed by that of VarILR_1 of SBP1 and SBP2 (Table 3). The variance of the second component of ILR (VarILR_2) is opposite to that of VarILR_1. The COV and corresponding

CC followed the same law of SBP1 > SBP3 > SBP2. The first ILR equation ($z_1$ in Table 1) contained all the information of soil PSFs, while the second ILR equation ($z_2$ in Table 1) included only two components. The information of VarILR_1

was therefore more abundant. All  VarILR_1 and VarILR_2 values were not 0 (or not nearly 0), indicating that there was no constant (or almost constant) value in any two ratios of soil PSF components. The COV of SBP3 was close to 0, indicating that the proportions of *clay/sand* and *clay/silt* were approximately the same. The same results were generated from the corresponding CC. For the distribution of soil PSFs in a ternary diagram (the United States Department of Agriculture texture triangle, USDA), the main texture class was silt loam (Fig. 3a). A biplot of soil samples demonstrated that the rays of the three components, i.e., sand, silt, and clay, were reasonably well clustered at about 120° in the three groups (Fig. 3b).

**Table 3.** Covariance structure of soil PSFs based on different SBPs. VarILR_1 and VarILR_2 denote the variance of the first and the second components of ILR, respectively. COV refers to the covariance of ILR1 and ILR2. CC is the correlation coefficient.

| Balances | VarILR_1 | VarILR_2 | COV | CC |
|---|---|---|---|---|
| SBP1 | 0.53 | 0.71 | 0.32 | 0.52 |
| SBP2 | 0.39 | 0.86 | -0.24 | -0.41 |
| SBP3 | 0.94 | 0.30 | -0.09 | -0.16 |

[Figure]

**Figure 3.** Distribution of soil PSFs in the USDA triangle (a) and biplot graph (b). The red curve was fitted using a loess function.

**3.2 Comparison of the accuracy of different models using ILR data**

To assess the accuracy of the different models, the Diebold–Mariano test was used, which showed that the statistical differences of most models were significant. This significance was reflected not only in different models (GLM and RF), but also in different SBPs when using the same model (Tables S6.1–S6.7). The first three rows of the boxplots in Figs. 4a–4c indicate the bias of the different models according to their ME values. The ME

of sand was closest to 0, followed by the MEs of clay and silt. GLM was more unbiased than RF, with lower ME values. After combining with RK, there was a marked improvement in the ME for most GLM and RF models (Figs. 4a 4c).

With respect to the accuracy assessment, the RMSE of silt was higher than that of the other two components. The

GLMRK did not perform as well as expected in terms of the RMSE, with only the sand component having an improved RMSE

(Fig. 4d). However, the RFRK performed better than the GLMRK and improved the accuracy of most components
compared with the RF, except for the RFRK_SBP1 of sand. As an overall indicator, AD showed that the RF (or RFRK)
performed better than the GLM (or GLMRK) in terms of both average RMSE values and uncertainties (Fig. 4g). Moreover,
the RFRK improved the AD values for the SBP2 and SBP3 methods. With respect to the uncertainty assessment, the RF
generated lower uncertainties than the GLM according to bias, and the models combined with RK further reduced the
uncertainties, especially for the sand fractions of most  models (Table. S7.1).

[Figure]

**Figure 4.** Comparison of the accuracies of GLM, RF, and their RK patterns combined with three ILR
balances. The mean values of different model indicators were calculated in their boxes.

The model performances were different for the three SBPs. To better evaluate model performance using the different SBP

balances, we graded each box from 1 to 3,  based on the predicted results and Diebold–Mariano test results—the final results are shown in Fig. 5.  SBP1 performed best in terms of bias, with the lowest ME score of all models. , except for GLMRK (Fig. 5a). With respect to the model accuracy assessment, there was no apparent pattern, but the accuracy could be considered hierarchically: (1) for the GLM, SBP3 performed better than the other two SBP methods, and SBP1 performed well when  combined with RK (GLMRK); (2) for RF, SBP1

produced the best result. However, the introduction of RK resulted in the SBP2 and SBP3 performing well (Fig. 5b). In addition, SBP3 and SBP1

delivered a better performance for GLM and RF, respectively (Figs. 5c–5e). Finally, in a comprehensive assessment, SBP1 performed best out of the three SBPs according to SUM6 (Fig. 5f). More details and calculations can be found in the Supplementary Material (Table S4.1).

[Figure]

[Figure]

**Figure 5.** Ranking score of model performance based on three SBPs. SUM1 (a) and SUM2 (b) are the sum scores of ME and RMSE for each model, respectively. SUM3 (c) is the sum score of ME, RMSE, and AD for each model; SUM4 (d) and SUM5 (e) are the sum scores of ME or RMSE for GLM$_{all}$ (GLM and GLMRK) and RF$_{all}$ (RF and RFRK); SUM6 (f) is the sum score of all indicators. The lower the value of these scores, the better the model performance.

**3.3 Spatial prediction maps of soil PSFs generated from the different models**

Prediction maps of soil PSFs constructed from the different models are shown in Figs. 6, S3.1, and S3.2. For the components of soil PSFs, the prediction maps of the three components followed a similar rule. The GLM and GLMRK produced broader ranges of predicted values, and their maps were more relevant to the real environment. However, the RF and RFRK predicted a relatively narrow range of low values for these components (sand, silt and clay), revealing a smoother distribution than that generated by the GLM and GLMRK. Unlike the regression methods, the hybrid methods (GLMRK and RFRK  produced hot and cold spots on the prediction maps and more details of the soil sampling points were apparent (Fig. S5.1).

[Figure]

**Figure 6.** Spatial prediction maps of the sand component of the upper reaches of the Heihe River Basin.

**3.4 Spatial distribution of soil texture classes in the USDA triangles**

The predicted soil textures in the USDA texture triangles (Fig. 7) showed that most predictions fell within the range of observed soil textures (Fig. 3a), and silt loam was the dominant soil texture in all cases. The GLM produced a more discrete distribution than the RF, and the RK method expanded the dispersion. With respect to trends in the predicted samples, the silt components predicted from all models were overestimated. The pattern fitting curves indicated that the prediction results were closer to the bottom right of the USDA triangle than the soil PSF observations. The GLMRK and RFRK curves were longer than the GLM and RF curves, with a more extensive range of values in triangles. Compared with the GLMRK, the RFRK

produced a more upward extension (Figs. 7j, k, l 7l). It was clear that the clay fraction was overestimated and the sand fraction was underestimated.

[Figure]

**Figure 7.** Predicted 262 soil samples in USDA texture triangles using (a) GLM_SBP1, (b) GLM_SBP2, (c) GLM_SBP3, (d) GLMRK_SBP1, (e) GLMRK_SBP2, (f) GLMRK_SBP3, (g) RF_SBP1, (h) RF_SBP2, (i) RF_SBP3, (j) RFRK_SBP1, (k)

RFRK_SBP2,  (l) RFRK_SBP3. Red fitting curves show the trends.

**4 Discussion**

**4.1 Comparison of the GLM, RF, and RK patterns using ILR data**

We found  RF provided more accurate results, but with more bias than the GLM, and  RK method improved the performance in terms of bias for most models and  accuracy of the RF. Odeh et al. (1995) indicated that RK was superior to  linear models, such as MLR, which was reflected in the prediction results for sand in our study. Scarpone et al. (2016)

reported that as a hybrid interpolator, the RFRK outperformed the RF when making soil thickness predictions. We proved that

RFRK was also suitable for compositional data and improved model performance when combined with the ILR

transformation. In summary, the GLM and RF had both advantages and disadvantages when considering the trade-off between bias and accuracy.

The results of GLM and GLMRK should not depend on the  ILR basis , which has been proved by previous studies on the use of linear models and kriging for compositional data (Pawlowsky-Glahn et al, 2015). However, the

GLM model used the "glmStepAIC" algorithm (i.e., a stepwise regression) to select the best combination of environmental covariables for each ILR component (Table S2.1). Therefore, the variable inputs are different for these ILR data, and further impact the accuracy assessment and prediction maps. In addition, the difficulty with  GLM is the need for a back- transformation. There is a need to present results on the original untransformed scale after conducting the analysis on a transformed level, which may produce spurious results (Lane, 2002). In our study, we compared the means of ILR transformed data and the original data. We proved the feasibility of the ILR transformation method, especially for meeting the requirements of compositional data. However, the accuracy of the GLM still needs to be improved, which may be because the transformed data did not follow a normal distribution (Fig. 2). With respect to uncertainty, the uncertainty of bias for GLM is higher than that of RF, but the uncertainty of accuracy for GLM is lower. However, RF performed better in terms of accuracy assessment.

Therefore, the main concern was whether the introductions of RK could reduce the uncertainty of RF. With regard to the performances of RFRK and RF, adding RK was recommended in soil PSF interpolation combined with ILR transformed data.

In addition, the range of 95% prediction interval for different models (Figs. S8.1–8.6) demonstrated that the differences were very close. This may because the values of variance for ILR data were small, showing low uncertainty when using ILR

transformed data.

Although the RF had the advantage of prediction accuracy, the limited interpretability of the consequences made it difficult to modify the prediction bias— each tree from the model could not be examined individually (Grimm et al., 2008).

Moreover, the ILR transformation before modeling increased the difficulty of interpretation for not only the predicted values on the ILR scale but also the residuals. The back-transformation of the optimal estimate of log-ratio variables did not generate an optimal estimation of compositional data (Lark and Bishop, 2007), which should also be considered.

**4.2 Comparison of three SBPs of ILR transformation**

Regarding the three SBPs, the ME and RMSE performed better when using SBP1 for ILR transformed data, which may be interpreted as the distributions of the ILR1 and ILR2 of SBP1 being more symmetric (Fig. 2b). In contrast, the performance of SBP2 was worse than that of SBP1 and SBP3 because the ILR_1 component, including all the soil PSF information, was left-skewed (Fig. 2c). This result was especially apparent for the GLM and GLMRK, because the data in a linear model needs to be normally distributed (Lane, 2002).

The negligible difference among the three SBP balances revealed a triangular shape with a cluster at about 120° (Fig. 3b). This could be interpreted as the three soil PSFs having a mixed pattern, with each component dominated by the components in one cluster (Tolosana-Delgado et al., 2005). Although the silt component dominated the soil PSFs (Fig. 2a), sand and clay also played important roles in soil composition. Taking either the most abundant component of the compositional data as the denominator (Martins et al., 2016) or the first component of the permutations did not provide convincing evidence  that the model performs best. This is because using the most abundant component of the compositional data as the primary component of the alterations, i.e., SBP2, resulted in a relatively poor performance compared to the other SBPs. Thus, we recommend that the focus should be on data distribution. Furthermore, the choice of balance and combination of RK are also the key to improving model accuracy, as shown by the result of the RFRK-SBP3 model (Fig. 4).

**4.3 Limitations**

Firstly, the scope of this study is limited to independent modeling. Each ILR component was modeled separately, which may be suboptimal because the components cannot take into account the cross correlations among the ILR coordinates. However, the study has demonstrated the relation of the raw data (sand, silt, and clay) based on ILR transformation, and has confirmed that  currently used prediction models in this work are suitable. In our pervious study, we have used compositional kriging (CK) for the spatial prediction of soil PSFs (Wang and Shi, 2017), and the cross correlations of ILRs can be taken into account using CK. Although CK is optimal, it cannot take into account different balances of ILR, nor can it be combined with a hybrid interpolator (e.g., RK). Moreover, predicting each ILR component separately was a more suitable approach for the spatial prediction models currently used (such as the GLM and RF). Therefore, more alternative spatial prediction models combined with interpretation of ILR balances for compositional data should be considered in the future. For example, CK and high-accuracy surface modelling (HASM; Yue et al., 2007; Yue, 2011; Yue et al., 2016) can be applied for small scale study areas. For large scale study areas, multivariate RF (Segal and Xiao, 2011) can be combined with a log-ratio transformation and hybrid interpolation methods, enabling the cross correlations among ILR coordinates to be better interpreted.

Secondly, the weighting problem was not considered in this study because the ILR method can be qualified as an unweighted log-ratio transformation, giving all parts the same weight for both the definition of the total variance and the reduction of dimensions. This may enlarge the ratios generated from the rare component, which would dominate the analysis (Greenacre and Lewi, 2009). The pairwise log-ratio can be used to set weights by their proportions when there is no knowledge of the component measurement errors (Greenacre, 2019). Nevertheless, all three parts of the soil PSF data dominated the biplot diagram, without the influence of rare elements and with no redundancy; thus, none of the shortcomings mentioned above were apparent. Accuracy assessments using a pairwise log-ratio transformation require further study in the future.

**5 Conclusions**

We evaluated and compared the performances of the GLM and the RF and their hybrid patterns (i.e.,

GLMRK and RFRK) using different balances of ILR transformed data. The bias of the GLM was lower than that of the RF; however, the accuracy of the GLM was relatively low. More discrete distributions and broader ranges of prediction value distributions were produced from GLMs in the USDA soil texture triangles. In other words, different predicted data sets were generated from the use of the GLM and RF, with unbiased and inaccurate predictions for the GLM and biased and more accurate predictions for the RF.

The hybrid patterns, GLMRK and RFRK, were found to provide the best solutions because they produced a relatively high prediction accuracy and strong correlations with ECs, providing more details about the soil sampling points (hot spots and cold spots) compared with using the regression model only. However, the non-normal distribution of ILR

data and their residuals, and increased data transformation and inverse transformation  make models more difficult to interpret and improve.

The three SBP-based datasets generated different distributions; a statistical significance test proved that most models had significant differences in prediction accuracy using different SBPs. A ranking score was provided to demonstrate these differences, and compositional balance should be considered when mapping soil PSFs. However, no pattern was apparent, which could be explained by the angle of the biplot diagram with three rays of soil PSF

components clustered into three modes, and each part dominating its cluster. Using the most abundant component of the compositional data as the first component of the permutations was not considered the best choice for mapping soil PSFs because SBP2 delivered the worst performance. Thus, we recommend that the focus should be on data distribution.

This study provides a reference for the spatial simulation of soil PSFs combined with ECs at the regional scale, and for choosing the balances of ILR transformed data.

*Data Availability.* We did not use any new data and the data we used come from previously published sources. Soil particle- size fractions data is available through our previous studies (Wang and Shi, 2017, 2018). Moreover, it also can be visited on this website: http://data.tpdc.ac.cn/zh-hans/data/7f91d36d-8bbd-40d5-8eaf-7c035e742f40/ (Digital soil mapping dataset of soil texture (soil particle-size fractions) in the upstream of the Heihe river basin (2012-2016); last access: 4 July 2020). The meteorological data can be accessed through http://data.cma.cn/ (last access: 4 July 2020). Environmental covariates data of soil physical and chemical properties and categorical maps can be obtained through http://data.tpdc.ac.cn/zh-hans/ (last access: 4 July 2020), including saturated water content, field water holding capacity, wilt water content, saturated hydraulic conductivity data (http://data.tpdc.ac.cn/zh-hans/data/e977f5e8-972b-42a5-bffe-cd0195f3b42b/, Digital soil mapping dataset of hydrological parameters in the Heihe River Basin (2012); last access: 4 July 2020), and soil thickness data (http://data.tpdc.ac.cn/zh-hans/data/fc84083e-8c66-4a42-b729-4f19334d0d67/, Digital soil mapping dataset of soil depth in the Heihe River Basin (2012-2014); last access: 4 July 2020). DEM data set is provided by the Geospatial Data Cloud site, Computer Network Information Center, Chinese Academy of Sciences. (http://www.gscloud.cn, last access: 4 July 2020).

[revised manuscript text omitted]